# p53 wild-type colorectal cancer cells that express a fetal gene signature are associated with metastasis and poor prognosis

Laura Solé [1,2,9], Teresa Lobo-Jarne[1,2,9], Daniel Álvarez-Villanueva [1,3], Josune Alonso-Marañón[1], Yolanda Guillén[1], Marta Guix [4], Irene Sangrador[1], Catalina Rozalén[1], Anna Vert[1], Antonio Barbachano [2,5], Joan Lop[6], Marta Salido[6], Beatriz Bellosillo [2,6], Raquel García-Romero[1], Marta Garrido[1], Jessica González[1], María Martínez-Iniesta[3], Erika López-Arribillaga [1,7], Ramón Salazar[2,3], Clara Montagut[2,4], Ferrán Torres [8], Mar Iglesias[2,6], Toni Celià-Terrassa [1], Alberto Muñoz[2,5], Alberto Villanueva [3], Anna Bigas [1,2✉] & Lluís Espinosa [1,2✉]

Current therapy against colorectal cancer (CRC) is based on DNA-damaging agents that remain ineffective in a proportion of patients. Whether and how non-curative DNA damage-based treatment affects tumor cell behavior and patient outcome is primarily unstudied. Using CRC patient-derived organoids (PDO)s, we show that sublethal doses of chemotherapy (CT) does not select previously resistant tumor populations but induces a quiescent state specifically to *TP53* wildtype (WT) cancer cells, which is linked to the acquisition of a YAP1-dependent fetal phenotype. Cells displaying this phenotype exhibit high tumor-initiating and metastatic activity. Nuclear YAP1 and fetal traits are present in a proportion of tumors at diagnosis and predict poor prognosis in patients carrying *TP53* WT CRC tumors. We provide data indicating the higher efficacy of CT together with YAP1 inhibitors for eradication of therapy resistant *TP53* WT cancer cells. Together these results identify fetal conversion as a useful biomarker for patient prognosis and therapy prescription.

[1] Cancer Research Program, Institut Mar d'Investigacions Mèdiques, CIBERONC, Hospital del Mar, Barcelona 08003, Spain. [2] CIBERONC, Instituto de Salud Carlos III, Madrid, Spain. [3] Chemoresistance and Predictive Factors Group, Program Against Cancer Therapeutic Resistance (ProCURE), Catalan Institute of Oncology (ICO), Oncobell Program, Bellvitge Biomedical Research Institute (IDIBELL), L'Hospitalet del Llobregat, Barcelona, Spain. [4] Department of Oncology, Institut Mar d'Investigacions Mèdiques, CIBERONC, Universitat Pompeu Fabra, Barcelona 08003, Spain. [5] Department of Cancer Biology, Instituto de Investigaciones Biomédicas 'Alberto Sols', Spanish National Research Council (CSIC)-Autonomous University of Madrid (UAM) and IdiPAZ, Madrid, Spain. [6] Department of Pathology, Institut Mar d'Investigacions Mèdiques, Universitat Autònoma de Barcelona, Barcelona 08003, Spain. [7] Group of Biomedical Genomics, Institut de Recerca Biomedica (IRB), Barcelona 08028, Spain. [8] Biostatistics Unit, Medical School, Universitat Autònoma de Barcelona, Barcelona, Spain. [9] These authors contributed equally: Laura Solé, Teresa Lobo-Jarne. ✉email: abigas@imim.es; lespinosa@imim.es

Colorectal cancer (CRC) remains the second leading cause of cancer-related death, which highlights the need for novel therapies focused on the treatment of advanced disease. Treatment of localized CRC currently involves surgery, radiotherapy, and/or chemotherapy (CT) (mainly 5-FU or capecitabine and oxaliplatin in the neoadjuvant or adjuvant setting), while CT (5-FU, oxaliplatin, and irinotecan) still represents the main backbone of treatment for advanced CRC. In general, classical CT agents are designed to eradicate tumors by inducing DNA damage in highly proliferative cells leading to cell death (reviewed in ref. [1]). However, most tumors contain a variable proportion of quiescent cells, including cancer stem cells, that are refractory to these agents thus contributing to tumor relapse and metastasis[2]. Supporting this notion, the presence of intestinal stem cell (ISC) signatures in tumors is predictive of poor prognosis in patients[3]. In addition, although current therapeutic regimes are devised to expose tumor cells to the maximum doses tolerated by patients, it was demonstrated that altered vascularization in tumors led to heterogeneous drug delivery thus impairing the efficacy of CT[4,5]. Consequently, even after adequate treatment, ~25–30% of CRC patients in the less aggressive stage II tumors and up to 30–50% in stage III relapse, and most of them eventually die (data from the American Cancer Society).

To avoid therapeutic resistance, strategies potentiating the effect of DNA-damaging agents are recurrently proposed as the base for more effective combination therapies[6,7]. Sublethal CT has also been suggested as an alternative treatment based on its ability to impose a senescent phenotype on cancer cells, characterized by high levels of the cell cycle inhibitors p16 and p21, cessation of proliferation, and presence of a senescent-associated-secretory-phenotype, which can delay disease progression[8] but also provide pro-tumorigenic factors to neighboring tumor populations (reviewed in refs. [9–11]).

Recent studies in glioblastoma and squamous cell carcinoma indicated that TGFβ or kinase inhibitor therapies can increase drug resistance by imposing a reversible-quiescent state on cancer cells[12,13]. Similarly, using CRC xenograft models, it was demonstrated that tumor cells that persist after CT display a phenotype resembling the quiescent/slow-cycling embryonic diapause-like stage[14]. YAP1 pathway, which drives fetal conversion in the regenerating intestine, has consistently been recognized as a tumor promoter and inducer of CT resistance in cancer[15]. However, recent work (using the Apc−/−; KrasG12D; p53−/− murine CRC model) indicated that fetal reprogramming induced by YAP1 led to tumor and metastasis suppression[16]. Thus, to date there is no conclusive data to establish the impact of YAP1 and fetal/embryonic conversion in CRC, and whether sublethal CT can directly impose specific adjustments to cancer cells.

We here show that CRC patient-derived organoids (PDO)s and cell lines exposed to sublethal CT acquire a non-senescent quiescent-like phenotype that persists after DNA damage resolution, which is restricted to TP53 WT cells. Cells acquiring this persistent quiescent-like (PQL) phenotype display higher in vivo metastatic capacity and express a YAP1-dependent fetal signature that is also detected in a subset of untreated CRC tumors. The presence of this specific fetal signature, or detection of nuclear YAP1 in tumors, predicts poor disease outcomes at stages II and III in patients carrying tumors with WT TP53.

## Results

### Low-dose CT treatment induces a non-senescent quiescent-like phenotype to TP53 WT cancer cells in the absence of sustained DNA damage.
To investigate the mechanisms imposing therapy resistance in cancer patients, we treated CRC PDOs with serial dilutions of CT agents 5-FU + Iri. As expected, high 5-FU + Iri.

concentrations led to the eradication of most PDO cells independently of their mutational status. However, we were able to define in all cases the $IC_{20}$ and $IC_{30}$ as the 5-FU + Iri. doses that reduced cell viability by 20 and 30% after 72 h of treatment, which was specific for each PDO (Fig. 1A and Supplementary Table S1). Microscopy analysis of PDO5 (TP53 WT) treated at $IC_{20}$ and $IC_{30}$ did not reveal obvious signs of cell death, but we observed a dose-dependent growth arrest in all tested PDO TP53 WT and the hypomorphic PDO4 that continued for at least 2 weeks after drug washout (Fig. 1B, C and S1A). In contrast, the TP53 deficient PDO8, PDO10, PDO11, and PDO15 totally failed to reinitiate tumor growth after drug washout (Fig. S1A). Growth arrest in TP53 WT PDOs was associated with proliferation inhibition as determined by immunohistochemistry (IHC) analysis of ki67 (Fig. 1D and S1B) and reduced number of cells in S phase with accumulation in $G_0/G_1$ and $G_2/M$ (Fig. S1C), the latter probably corresponding to cells not undergoing cytokinesis[17,18]. By fluorescent in situ hybridization (FISH) and DAPI staining, we demonstrated the absence of polyploid or multinucleated cells following $IC_{30}$ treatment (Fig. S1D). We determined whether $IC_{20}$ and $IC_{30}$ treatments inflicted a senescent phenotype to the TP53 WT PDO5 cells by evaluation of senescence-associated (SA)-β-Galactosidase activity by flow cytometry (Fig. 1E) and IHC (Fig. S1E). Cells that persisted after $IC_{20}$ or $IC_{30}$ were not senescent, in contrast with cells treated at $IC_{60}$ for 72 h (Fig. 1E). Accordingly, the addition of the senolytic agent dasatinib[19] did not potentiate the growth inhibition imposed by $IC_{20}$ and $IC_{30}$ 5-FU + Iri. but enhanced the effect of $IC_{60}$ treatment (Fig. 1F). Moreover, we did not detect apoptotic cells in PDO5 after $IC_{30}$ treatment as determined by cleaved-caspase 3 (cCas3) staining (Fig. S1F) and Annexin V staining (Fig. S1G) that was robustly detected in the TP53 mutant PDO4 and PDO8 (Fig. S1H). Cell cycle arrest in the absence of apoptosis was observed not only in PDO5 but also in the TP53 WT PDO20 and PDO66 (Fig. S1I). We studied whether cell cycle arrest after $IC_{20}$ and $IC_{30}$ treatment was linked to sustained DNA damage. Comet assay (Fig. 1G) and Western Blot (WB) analysis of γH2A.X (Fig. 1H) in PDO5 revealed a dose-dependent accumulation of DNA damage starting at 1–3 h with a maximum at 24 h. DNA damage was undetectable at 72 h after $IC_{20}$ and $IC_{30}$ treatment, but clearly present in $IC_{60}$-treated PDO5 (Fig. 1H, I). In contrast, PDO4 and PDO8 cells carrying mutated TP53 exhibited high amounts of DNA damage following $IC_{20}$ and $IC_{30}$ 5-FU + Iri. treatment that lasted for at least 72 h (Fig. 1J). Further supporting the differential response to sublethal CT in terms of DNA repair according to TP53 status, we detected massive amounts of γH2A.X by WB analysis of TP53 mutant PDOs and CRC cell lines upon $IC_{20}$ treatment (Fig. S1J and S1K).

In most TP53 mutant PDO cells included in our study, IC20-30 usually requires a higher concentration of 5-FU + Iri., which could contribute to the higher DNA damage that is produced. To test this possibility, we used two different paired models with comparable sensitivity to 5-FU + Iri. (see Table S1); PDO5 WT and TP53 KO (generated by CRISPR-Cas9) (Fig. 1K, L) and the HCT116 (TP53 WT) and DLD1 CRC (TP53 mutated) cell lines (Fig. 1M). We confirmed a DNA repair defective phenotype in p53 deficient cells treated at the same 5-FU + Iri. doses.

These results indicate that cancer cells carrying functional p53 respond to low CT by acquiring a quiescent-like phenotype, hereafter referred to as PQL, in the absence of sustained DNA damage.

### PQL cancer cells display increased in vitro and in vivo tumor initiation capacity.
We studied whether PQL cells preserved comparable tumor-initiating capacity (TIC) as untreated cancer

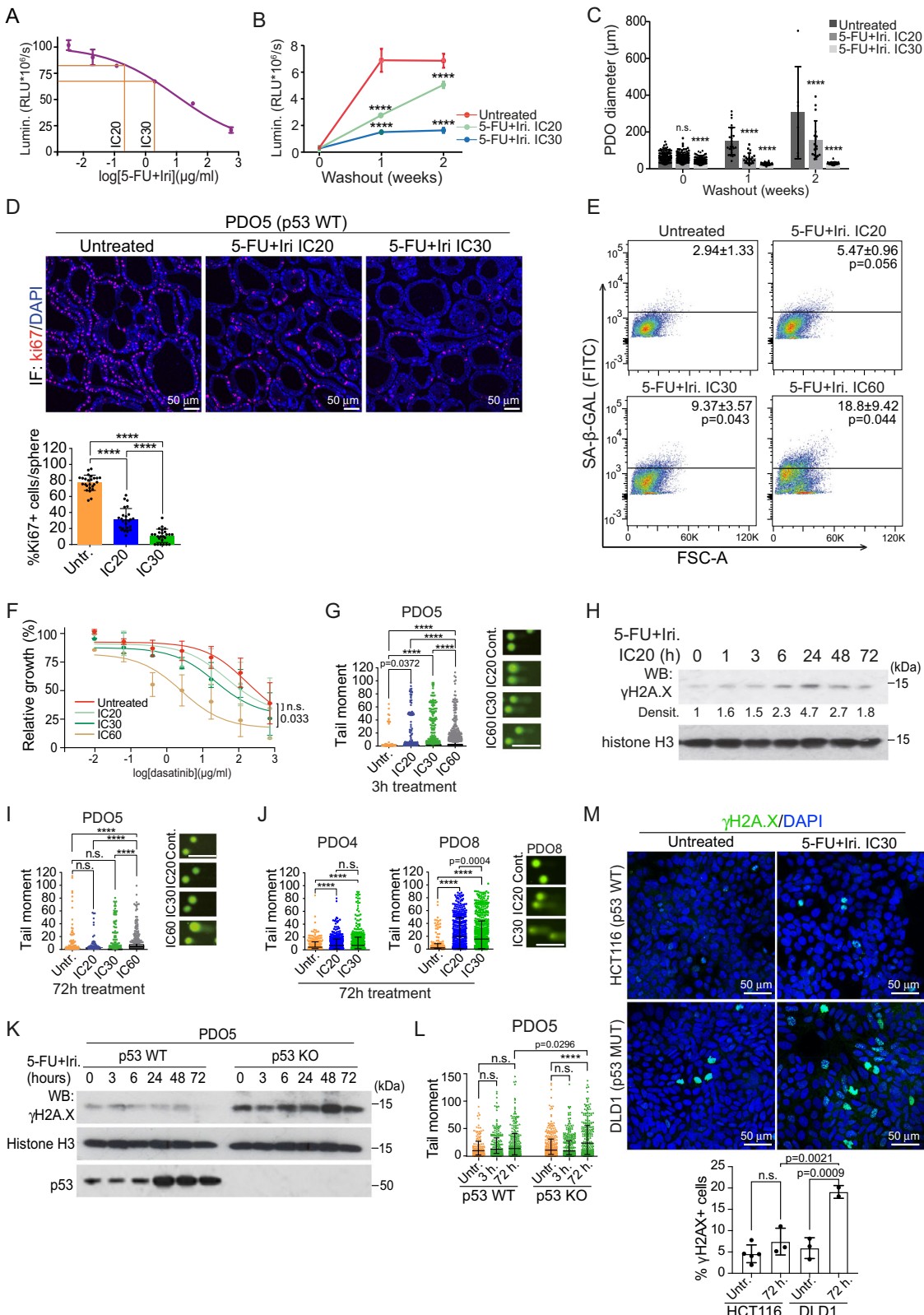

cells. We seeded 300 single cells from untreated or 5-FU + Iri. $IC_{20}$ or $IC_{30}$-treated *TP53* WT PDO5 and PDO66. We found that low-dose CT treatment of PDO5, PDO20, and PDO66 cells did not preclude TIC in vitro, as indicated by the slight reduction in the number of spheres generated compared to controls (Fig. 2A and S2A, upper panels), but imposed a reduction of spheres diameter, more pronounced in PDO5 (Fig. 2A and S2A, lower

panels), consistent with their low proliferation rates. In contrast, 5-FU + Iri. pre-treatment ($IC_{20}$) of *TP53* mutant PDO4 and PDO8 cells resulted in TIC abrogation (Fig. 2B). Similar results were obtained by KO of *TP53* in PDO5 (Fig. 2C). Considering that TIC activity in *TP53* WT PDOs could be driven by the fraction of cells that still undergo replication after $IC_{20}$ and $IC_{30}$ treatment (Fig. 1D, S1B and S1C), we next compared the TIC

**Fig. 1 Low-dose CT treatment induces a non-senescent quiescent-like state to CRC PDO in the absence of persistent DNA damage. A** Dose-response assay of PDO5 treated with 5-FU + Iri. for 72 h, indicating the $IC_{20}$ and $IC_{30}$ doses ($n = 3$ replicates examined, from one out of four biologically independent experiments). **B, C** Quantification of PDO5 **B** viability ($n = 3$ replicates examined, from 1 out of 4 biologically independent experiments) and **C** diameter ($n = $ at least 15 spheres examined whenever possible over four biologically independent experiments), after 72 h of 5-FU + Iri. treatment and 1–2 weeks of washout. **D** Representative ki67 stainings of PDO5 treated for 72 h as indicated, and quantification of ki67$^+$ cells/sphere ($n = 25$ spheres examined over 3 biologically independent experiments). **E** Flow cytometry analysis of SA-β-Gal activity in PDO5 treated as in **D**. **F** Dose–response curves of PDO5 treated with dasatinib for 3 days after 5-FU + Iri. pre-treatment, as indicated ($n = 2$–3 biologically independent experiments). **G** Comet assay in PDO5 treated for 3 h as indicated ($n = $ more than 700 cells examined over three independent experiments). **H** WB analysis of the DNA damage sensor γH2A.X in PDO5 cells collected at the indicated time points after 5-FU + Iri treatment (from one out of three biologically independent experiments). **I, J** Comet assay in 53 WT PDO5 **I** and p53 mutants PDO4 and PDO8 **J**, treated for 72 h as indicated ($n = $ >800 cells examined over 3 independent experiments). **K, L** WB analysis **K** (from one out of three biologically independent experiments) and comet assay **L** of PDO5 CT and *TP53* KO untreated or treated with 5-FU + Iri. at the same concentration at the indicated time points ($n = $ more than 490 cells examined over 3 independent experiments). **M** Representative γH2A.X staining and quantification of HCT116 and DLD1 cell lines treated for 72 h with 5-FU + Iri. at the same concentration (5-FU 0.1 μg/mL and Iri. 0.04 μg/mL) ($n = $ more than 50 cells examined over 2-3 biologically independent experiments). For all applicable figure panels, data are mean ± SD, except for **G, I, J**, and **L** (Tukey method for box plots), where boxes represent the central 50% of the data (from the lower 25th percentile to the upper 75th percentile), lines inside boxes represent the median (50th percentile), and whiskers are extended to the largest value less than the sum of the 75th percentile plus 1.5 IQR (the difference between the 25th and 75th percentile) or greater than the 25th percentile minus 1.5 IQR, and plot any values that are greater or lower than this as individual points. Significance (p) was calculated with one-way ANOVA test, except for B and C (two-way ANOVA) and F (two-sided logistic regression trend test). For **G, I**, and **J** scale bar represents 50 μm. ****$p < 0.0001$; n.s., no significant. 5-FU, 5-fluorouracil; Iri, irinotecan; SA-β-Gal, SA-β-Galactosidase; $IC_{20}$, $IC_{30}$, and $IC_{60}$ indicate 5-FU + Iri. treatment leading to 20, 30, and 60% cell death, respectively. Source data are provided as a Source Data file.

in vitro of the general $IC_{20}$-treated PDO5 population and purified quiescent cells. For this, we generated a PDO5 line (PDO5-hG) carrying a doxycycline-inducible histone H2B-GFP reporter that is specifically retained by the quiescent tumor population after doxycycline withdrawal[20]. Upon 6 days of doxycycline treatment, PDO5-hG cells were treated with 5-FU + Iri. for 72 h and, after 2 weeks of doxycycline washout, analyzed by flow cytometry and $GFP_{high}$ and $GFP_{low}$ were sorted (Fig. S2B). We found that sorted $GFP_{high}$, which represents the quiescent population of CT-treated cells, displayed an identical capacity for an organoid generation as $GFP_{high}$ plus $GFP_{low}$ cells (Fig. S2C) indicating that TIC activity is retained in the PQL population. By transplantation of 5000 $GFP_{high}$ or $GFP_{low}$ PDO5 sorted cells in the cecum of nude mice, we confirmed the higher clonogenic activity of the quiescent population in vivo. In particular, 3 out of 5 mice transplanted with $GFP_{high}$ cells developed tumors in the cecum and/or intraperitoneal implants compared with 1 out of 5 mice transplanted with $GFP_{low}$ cells after 2 months (Fig. S2D).

We next studied the in vivo tumorigenic and metastatic capacity of $IC_{20}$ and $IC_{30}$-pre-treated PDO5 cells using two complementary strategies. First, we performed intracardiac injection of 40,000 single cells (untreated, $IC_{20}$ or $IC_{30}$ pre-treated) labeled with firefly luciferase into NOD-SCID-gamma (NSG) immunocompromised mice. Mice were analyzed weekly using bioluminescence to monitor metastatic growth using the IVIS animal imaging system (Fig. 2D). We found that PDO5 treated with 5-FU + Iri. displayed a superior and dose-dependent, although non-significant, metastatic capacity than untreated cells (logistic regression trend test, $p = 0.108$). Specifically, 7 of 14 mice transplanted with untreated PDO5 cells showed visible metastasis 15 weeks after injection compared with 4 of 6 mice transplanted with $IC_{20}$-treated cells and 9 of 11 mice with $IC_{30}$-treated cells (Fig. 2D, E). Quantitative analysis of the evolution of lesions in an independent assay demonstrated a significantly higher capacity of $IC_{30}$-treated cells for metastasis initiation (Fig. 2F).

Next, we inoculated equivalent numbers of untreated, $IC_{20}$ and $IC_{30}$ pre-treated PDO5 cells in the cecum of athymic nude mice. Tumor growth was assessed by palpation weekly and animals were sacrificed synchronously 70 days after transplantation. We found that untreated, $IC_{20}$ and $IC_{30}$-treated PDOs all generated tumors at the site of inoculation, being $IC_{20}$ and $IC_{30}$-treated derived tumors significantly smaller than those arising from

untreated controls (Fig. 2G), as expected. Importantly $IC_{20}$ and $IC_{30}$-treated PDO cells displayed a significantly higher ability to generate intraperitoneal implants when compared with untreated tumor cells (Fig. 2G–I). Still, we detected a reduction in the proliferation capacity of CT-treated PDO5 cells as determined by IHC analysis of the proliferation marker ki67 (Fig. 2J, K). Parallel in vivo experiments comparing $IC_{20}$-treated PDO5, PDO4, and PDO8 cells indicated a defective capacity of p53 mutant cells to generate in situ tumors and intraperitoneal implants after sublethal CT treatment (Fig. 2L), which was in agreement with their defective TIC in the in vitro assays.

These results indicate that *TP53* WT PDO5 cells show reduced capacity to proliferate in vitro and in the primary tumors after treatment with sublethal 5-FU + Iri. but comparable TIC as untreated cells in vitro and higher metastatic activity in vivo.

**CT-induced PQL cells display fetal intestinal stem cells (feISC) characteristics.** Sublethal CT treatment has been linked to the acquisition of specific stem cell signatures in B-cell lymphoma[10] and intestinal cancer[21]. To study the transcriptional changes associated with the PQL phenotype, we performed RNA sequencing (RNA-seq) of control, $IC_{20}$- and $IC_{30}$-treated PDO5, and $IC_{30}$-treated PDO66 cells. Bioinformatic examination of data demonstrated a significant overlap between differentially expressed genes (DEG) in PDO5 and PDO66 (Supplementary Data 1). Moreover, DEG in $IC_{20}$- and $IC_{30}$-treated PDO5 showed an almost perfect correlation of gene expression in pairwise comparisons ($IC_{20}$ vs. untreated and $IC_{30}$ vs. untreated) ($p < 2.2e$–16, $R = 0.974$) (Fig. S3A). Principal Component Analysis (PCA) indicated that untreated PDO5 and PDO66 clustered together and formed separate entities when compared with IC20- and IC30-treated PDO5 or PDO66 cells (Fig. 3A). Gene Set Enrichment Analysis (GSEA) of genes differentially expressed in CT-treated PDO5 (Fig. 3B) and PDO66 (Fig. 3C) uncovered p53 as the main activated pathway in CT-treated cells, which was confirmed in PDO5 by WB analysis (Fig. 3D), qPCR (Fig. 3E) and ChIP assay (Fig. S3B) of canonical p53 targets. DEGs genes are also clustered in the NF-κB, EMT, and the interferon-gamma (IFNγ) pathway (Fig. 3B, C), which has been associated with inflammatory response and stemness[22–26]. Conversely, the E2F pathway that coordinates cell cycle progression at the G1/S transition (reviewed in ref. [27]) and the G2/M checkpoint were

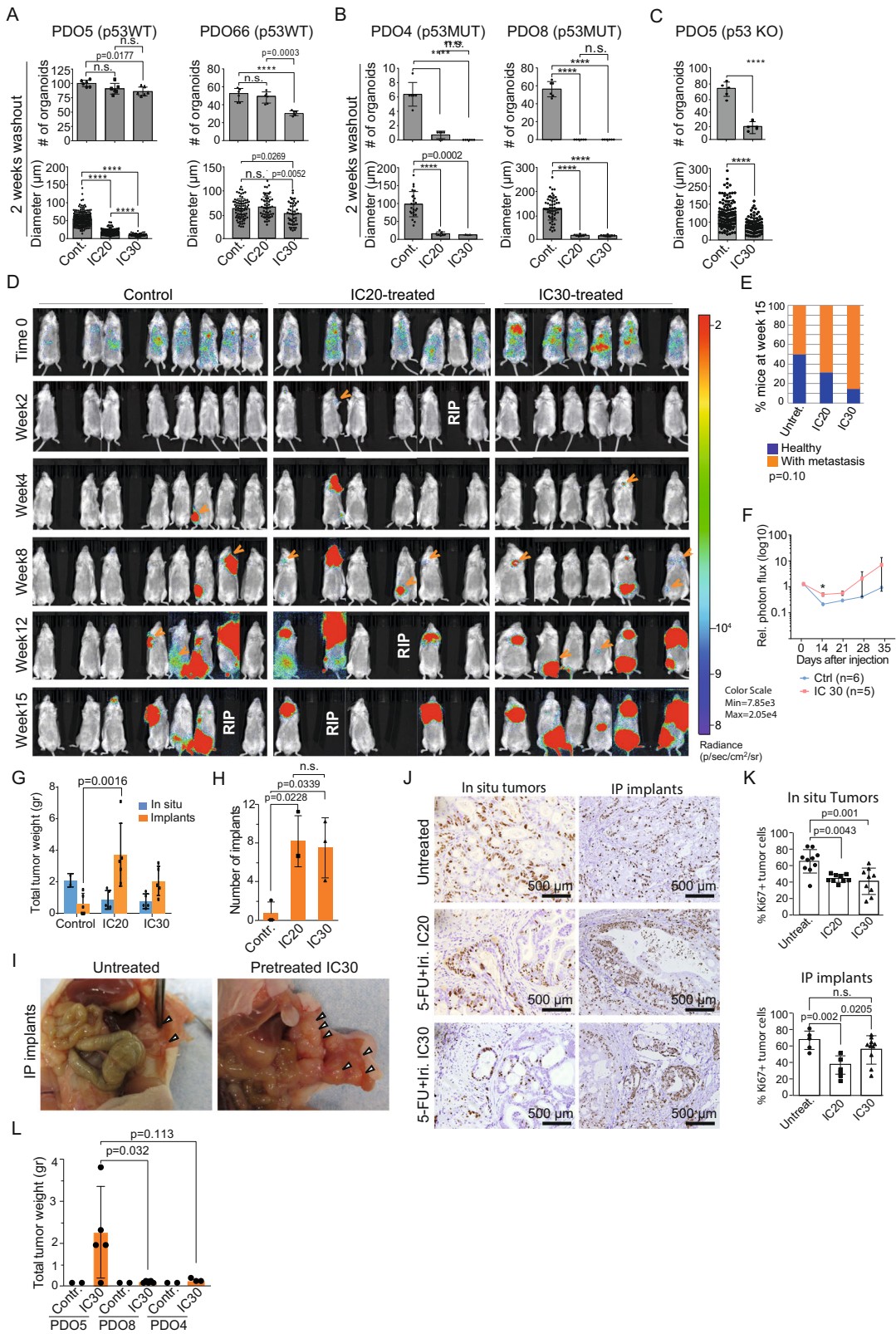

among the highest downregulated pathways, together with the MYC pathway (Fig. 3B, C), suggesting a general inhibition of proliferation.

Unexpectedly, our analysis identified an inversed correlation between DEG in PDO5 and PDO66 and the canonical Lgr5+ ISC signature[28] (Fig. 3F). More in-depth analysis showed a mixed pattern of genes upregulated such as *LY6D* and *YAP1*, which

are instrumental in the fetal ISC (feISC) after intestinal injury[16,26,29,30], and downregulated such in the case of canonical adult ISC markers *LGR5* and *EPHB2* (Fig. 3G). Accordingly, GSEA indicated a significant correlation between the CT-induced signature and the transcriptional program associated with fetal ISC conversion and loss of adult Lgr5+ cells[30] (Fig. 3H, I), which was found to be YAP1-dependent in the regenerating

**Fig. 2 _TP53_ WT PQL cells retain tumor-initiating capacity in vitro and in vivo. A–C** Number of PDOs (upper panels) and diameter (lower panels) of _TP53_ WT PDO5 and PDO66 **A**, _TP53_ mutant PDO4 and PDO8 **B**, and _TP53_ KO PDO5 **C** treated with 5-FU + Iri. as indicated, and left for 2 weeks with fresh medium. 300 cells/well were seeded ($n = 6$ wells examined for TICs and $n = $ more than 50 spheres examined whenever possible for diameters, from three biologically independent experiments). **D** Bioluminescence images of NGS mice after intracardiac injection of 40,000 luciferase-PDO5 CT and $IC_{20}$ or $IC_{30}$-treated cells (from 1 out of 2 biologically independent experiments). **E** Percentage of healthy and metastasis-carrying mice at week 15 ($n = 14$ (Untreat.), 6 ($IC_{20}$), and 11 ($IC_{30}$) mice examined over two biologically independent experiments). **F** Relative photon flux measurement of metastasis initiation in mice injected with PDO5 CT and $IC_{30}$-treated cells ($n = 6$ (Untreat.) and 5 ($IC_{30}$) mice examined from one experiment). **G–I** Total tumor weight of in situ tumors and intraperitoneal implants per animal in the different experimental groups **G** ($n = 5$ mice examined over two biologically independent experiments), number of intraperitoneal implants **H** ($n = 3$ mice examined from 1 experiment), and photographs of tumors derived from orthotopically implanted CT, $IC_{20}$ and $IC_{30}$-pre-treated PDOs in nude mice **I**. **J, K** IHC analysis of ki67 in in situ tumors and implants **J** and percentage of ki67$^+$ cells in the indicated conditions **K** ($n = $ more than four independent regions examined). **L** Tumor weight of intraperitoneal implants of tumors derived from orthotopically implanted CT and $IC_{30}$-pre-treated PDO5, PDO8 and PDO4 in nude mice ($n = 2$ (Untreat.) and 5 ($IC_{30}$) mice examined from 1 experiment). For all applicable figure panels, data are mean ± SD. Significance ($p$) was calculated with one-way ANOVA test, except for **C** (two-sided Student's _T_ test), **E** (two-sided logistic regression trend test) and **F** and **G** (two-way ANOVA test). ****$p < 0.0001$; n.s., no significant. 5-FU, 5-fluorouracil; Iri, irinotecan. Source data are provided as a Source Data file.

intestine[29,31]. We used the PDO5-hG line to test whether upregulation of genes contributing to feISC and EMT pathways upon sublethal CT was present in the quiescent cell population. Cells were treated as explained before (see Fig. S2B), sorted based on GFP levels, and processed for qPCR analysis. We detected a massive upregulation of feISC and EMT genes in CT-treated cells that were restricted to the GFP$_{high}$ population (Fig. 3J).

**Acquisition of feISC by CT treatment is linked to and dependent on YAP1 activation.** We investigated whether feISC conversion of human CRC cells induced by CT was imposed by YAP1 signaling. By WB analysis of PDO5 cells, we detected increased YAP1 expression after 72 h of 5-FU + Iri. treatment (Fig. 4A). In addition, we detected an accumulation of nuclear (active) YAP1 in $IC_{20}$ and $IC_{30}$-derived PDO5 tumors 2 months after implantation in mice (Fig. 4B). We studied whether YAP1 activity was required for transcriptional induction of feISC genes in PDO cells after CT. Incubation of PDO5 cells with the YAP1 inhibitor verteporfin precluded induction in several randomly selected fetal genes following $IC_{20}$ 5-FU + Iri. treatment (Fig. 4C). Indicating the specificity of verteporfin effects, a PDO5 YAP1 KO pool generated by CRISPR-Cas9 (Fig. S4A) showed comparable impairment of feISC genes induction by CT (Fig. 4D).

To study whether a similar mechanism operates in patients, we took advantage of a set of paired CRC samples ($n = 62$) collected at diagnosis (biopsy) and at the time of surgery ($n = 62$) from tumors showing partial response after DNA-damaging-based neoadjuvant treatment (Supplementary Table S2). By IHC analysis, we detected nuclear YAP1 in the epithelial component of 10 out of 46 tumors at diagnosis, considering positive those carrying ≥20% positive cells. Notably, the number of tumors carrying nuclear YAP1 was massively increased after neoadjuvant treatment (35 out of 45 analyzed) (Fig. 4E and Supplementary Table S2), which was associated with the expression of the feISC markers S100A4 and SERPINH1 (Fig. 4F). Neoadjuvant treatment was also linked to a general reduction in tumor cell proliferation, as determined by IHC analysis of ki67, in the absence of senescence traits such as enlarged nuclei (Supplementary Table S2) or high p16 levels (Fig. S4B). These characteristics were suggestive of human CRC tumors experiencing PQL conversion after sublethal CT. However, the limited number of samples in this cohort and the fact that a majority of post-treatment samples displayed nuclear YAP1 accumulation and proliferation inhibition precluded establishing the potential prognosis value of these parameters.

Since nuclear YAP1 was also detected in untreated tumors, we performed IHC analysis of YAP1 (Fig. 4G) in a tissue microarray containing 196 different human CRC samples (in triplicates)

obtained at diagnosis with available clinical data (Supplementary Table S6). We determined the H-score of nuclear YAP1 as intensity multiplied by the percent of positive tumor cells in the triplicates, and stratified patients accordingly. Considering the mean value ± 0.2 standard deviations of the H-score, we observed a trend towards poor prognosis in the group with higher nuclear YAP1 (disease-free survival: $p = 0.26$; HR = 1.38) that increased when considering mean value ± 0.4 s.d. ($p = 0.12$; HR = 1.58). Prognosis value of nuclear YAP1 levels reached statistical significance when considering the mean value ± 0.6 standard deviations ($p = 0.039$; HR = 1.97) (Fig. 4H, I). Indicative of the clinical applicability of our observations, verteporfin treatment increased the sensitivity of PDO5 cells to 5-FU + Iri. (Fig. 4J). Further suggesting that verteporfin effects are linked to YAP1 inhibition, genetic YAP1 deletion precluded PDO5 clonogenicity in TIC assays (Fig. 4K) associated with increased basal and CT-induced DNA damage (Fig. 4L).

Together these results indicate that feISC conversion imposed by CT is YAP1 dependent. Moreover, detection of nuclear YAP1 in CRC tumors is predictive of poor prognosis, which could be therapeutically exploited.

**A restricted YAP1-dependent fetal signature shows coordinate expression in human CRC associated with higher p21 levels.** The presence of nuclear YAP1 in untreated tumors associated with poor prognosis, led us to speculate that a YAP1-dependent fetal signature could already be present in tumors at diagnosis. We took advantage of public data sets to interrogate the incidence and prognosis value of feISC signature in untreated CRC human tumors. Computational analysis of the Marisa[32] (GSE39582), Jorissen[33] (GSE14333) and TCGA (TCGA Portal) CRC data sets using CANCERTOOL[34] indicated differential expression of most feISC genes with some of them distributed in clusters of coordinated expression (with either positive or negative correlation) (Supplementary Data 2, in the Marisa data set). We integrated feISC genes with the highest coordinated expression in a new cluster containing 28 plus 8 genes that were either upregulated (28up) or downregulated (8down) in CT-treated PDO cells and fetal ISCs (Fig. S5A). The 28up + 8down-feISC gene signature was present in Marisa (Fig. 5A), Jorissen, and TCGA CRC cohorts (Fig. S5B) associated with higher levels of the cell cycle regulator p21 (_CDKN1A_ gene) (Fig. S5C), further suggesting a link between fetal traits and slow-cycling phenotype. We confirmed upregulation of several genes of the 28up + 8down-feISC signature by RT-qPCR (Fig. 5B) and WB analysis (Fig. 5C) of $IC_{20}$ 5-FU + Iri.-treated PDO5. Activation of fetal genes following CT treatment was comparably observed in PDO66 (Fig. 5D) and slightly reduced in the hypomorphic

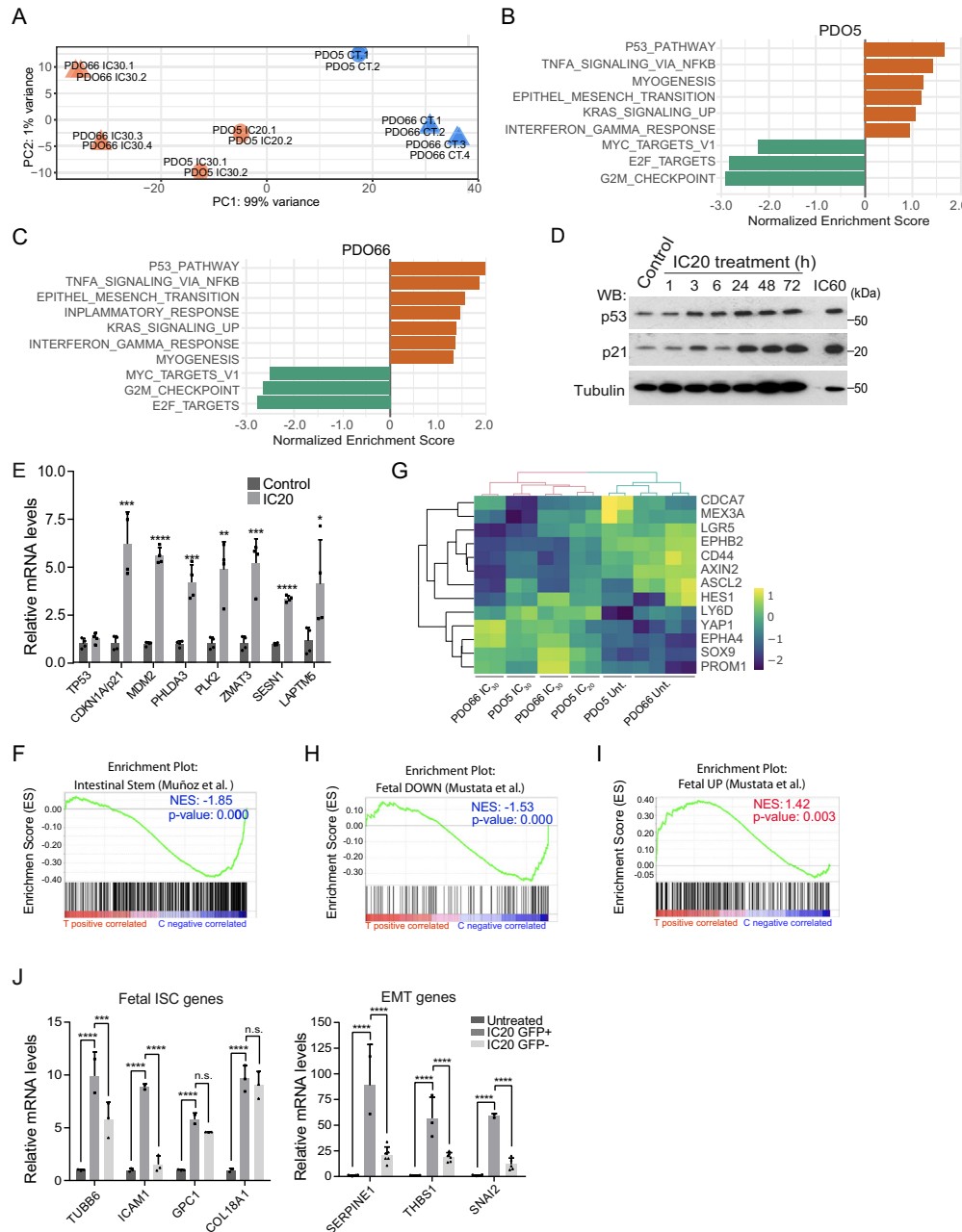

**Fig. 3 PQL phenotype is associated with the acquisition of a fetal intestinal stem cell (feISC) signature. A** PCA of the RNA-seq analysis from the indicated replicates performed on PDO5 and PDO66 (untreated; CT) or 5-FU + Iri.-treated either at IC20 or IC30 as indicated. **B**, **C** Barplots depicting the normalized enrichment score of statistically significant enriched pathways obtained by GSEA analysis of PDO5 (**B**) and PDO66 (**C**) with the Hallmark gene set for treated samples (NOM p val < 0.05). **D** WB analysis of control and 5-FU + Iri.-treated PDO5 cells collected at the indicated time points (from one out of three biologically independent experiments). **E** RT-qPCR analysis of selected p53 target genes from control and $IC_{20}$-treated PDO5 cells ($n = 4$ biologically independent experiments). **F** GSEA of an intestinal stem cell (ISC) gene set, according to Muñoz et al., in untreated (C) versus treated (T) PDO5/PDO66 condition. **G** Heat map showing the expression levels of the indicated ISC genes in untreated, $IC_{20}$ and $IC_{30}$-treated PDO5 cells and untreated and $IC_{30}$-treated PDO66 cells. **H**, **I** GSEA of a fetal down (**H**) and a fetal up (**I**) stem cell gene set, according to Mustata et al., in control (C) versus treated (T) PDO5/PDO66 condition. **J** RT-qPCR analysis of selected fetal ISC and EMT genes from untreated and $IC_{20}$-treated sorted $GFP_{high}$ or $GFP_{low}$ PDO5 cells (from 1 out of 2 biologically independent experiments). For all applicable figure panels, data are mean ± SD. Significance (p) was calculated in E with two-sided Student's t test and in **J** with one-way ANOVA test. For **F**, **H**, **I** p.value is nominal p value given by the GSEA program. *$p < 0.05$; **$p < 0.01$; ***$p < 0.001$; ****$p < 0.0001$; n.s., no significant; CT, control; $IC_{20}$ and $IC_{30}$, 5-FU + Iri. are doses imposing 20 and 30% cell death, respectively; GSEA, gene set enrichment analysis; NES, normalized enriched score. Source data are provided as a Source Data file.

TP53 mutant PDO4 (Fig. 5E) and the TP53 KO PDO5 pool (Fig. 5F). At the protein level, we detected slight differences in the activation of several feISC markers (such as TIMP2, MRAS, ICAM, or TSPAN4) after 5-FU + Iri. treatment when comparing TP53 WT and KO PDO5 (Fig. 5G) or CRC cell lines

carrying WT or mutant TP53 (Fig. 5H) but the massive accumulation of apoptotic cells as determined by cCas3 and cPARP1 levels (Fig. 5G, S1H, and S1I). By ChIP-seq assay of 5-FU + Iri. $IC_{20}$-treated PDO5 cells, we only detected 3 genes in the 28up-feISC signature, PLK2, PHLDA3 and GSN, that were

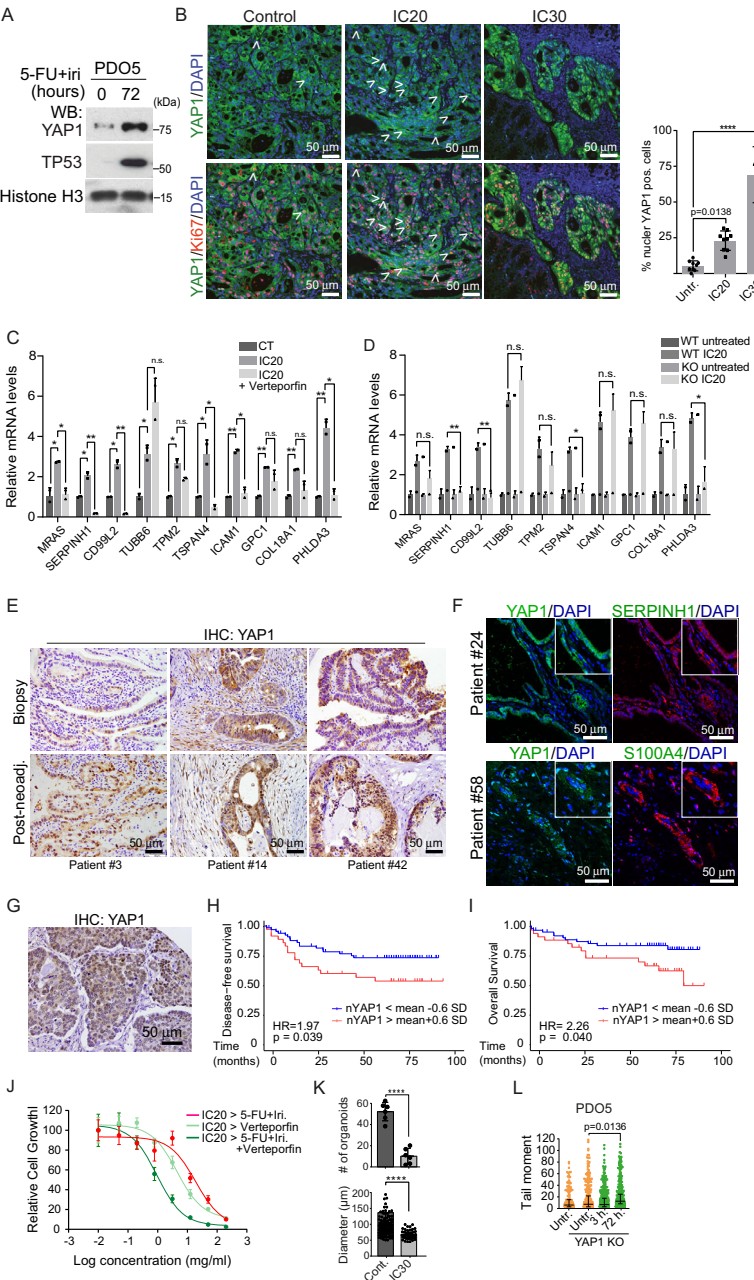

**Fig. 4 Acquisition of feISC by CT treatment is dependent on YAP1 activation. A** WB analysis of PDO5 cells was collected after 72 h of 5-FU + Iri. treatment (from one out of three biologically independent experiments). **B** Representative images and quantification of YAP1 and ki67 staining in tumors derived from control (CT), $IC_{20}$, and $IC_{30}$-pre-treated PDOs implanted in nude mice. White arrows indicate cells with nuclear YAP1. DAPI was used as a nuclear marker ($n = 9$ independent regions examined). **C** RT-qPCR analysis of randomly selected feISC genes in control and 5-FU + Iri-treated PDO5 alone or plus 0.2 μM verteporfin ($n = 2$ biologically independent experiments). **D** RT-qPCR analysis of same genes in control and *YAP1* KO PDO5 treated as in C ($n = 2$ biologically independent experiments). **E, F** Representative images of YAP1 **E** and SERPINH1 and S100A4 **F** in the indicated tumors at diagnosis or after neoadjuvant treatment (from 1 out of 6 biologically independent samples). **G** Representative image of an untreated human CRC identified as nuclear YAP1 high (done with all TMA samples, described in Supplementary Table S6). **H, I** Kaplan–Meier representation of disease-free **H** and overall **I** survival of CRC patients classified according to the H-score of nuclear YAP1. **J** Dose–response curves of $IC_{20}$-pre-treated PDO5 and then treated for 3 days as indicated ($n = 3$ replicates examined, from one out of three biologically independent experiments). **K** Number of PDOs (upper panel) and diameter (lower panel) of PDO5 *YAP1* KO after 72 hours of 5-FU + Iri. treatment and 2 weeks of washout. 300 cells/well were seeded ($n = 6$ wells examined for TICs and $n =$ more than 50 spheres examined whenever possible for diameters, from three biologically independent experiments). **L** Comet assay in PDO5 *YAP1* KO treated with 5-FU + iri. for 3 and 72 hours as indicated ($n = >790$ cells examined over three biologically independent experiments). For all applicable figure panels, data are mean ± SD, except for L (Tukey method for box plots), where boxes represent the central 50% of the data (from the lower 25th percentile to the upper 75th percentile), lines inside boxes represent the median (50th percentile), and whiskers are extended to the largest value less than the sum of the 75th percentile plus 1.5 IQR (the difference between the 25th and 75th percentile) or greater than the 25th percentile minus 1.5 IQR, and plot any values that are greater or lower than this as individual points. Significance (p) was calculated with one-way ANOVA, except for D and K by two-sided Student's *t* test and for H and I by log-rank (Mantel–Cox) test. *$p < 0.05$, **$p < 0.01$, n.s., no significant. CT, control; $IC_{20}$ and $IC_{30}$, 5-FU + Iri. treatment leading to 20 and 30% cell death, respectively. Source data are provided as a Source Data file.

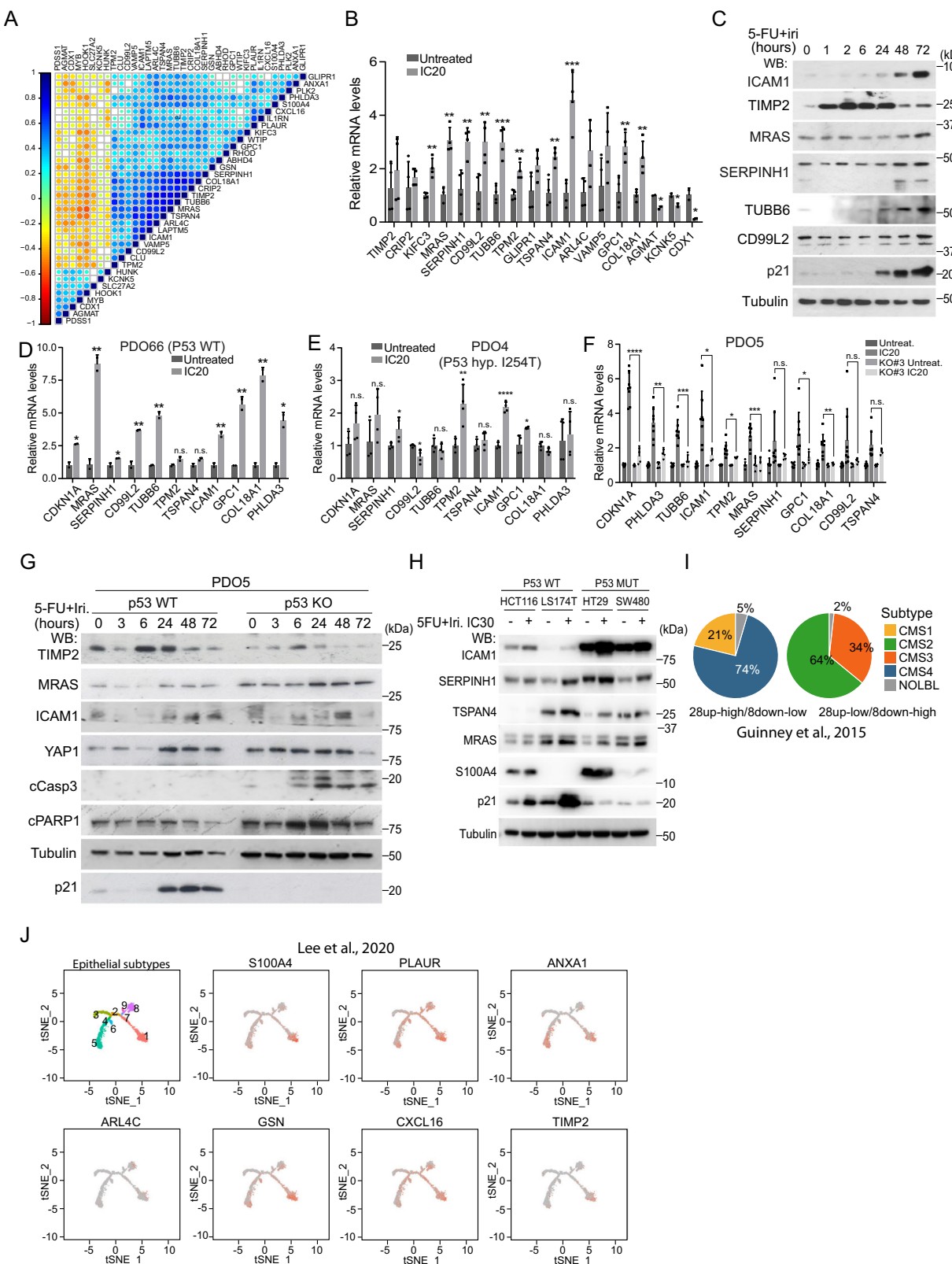

direct p53 targets (Fig. S5D). These results suggested that p53 does not participate in the transcriptional activation of feISC genes but might favor survival of cancer cells experiencing fetal conversion.

We also determined whether tumors carrying the 28up + 8 down-feISC signature were restricted to a specific cancer molecular subtype (CMS), based on the classification by Guinney

and collaborators[35]. 74% of tumors with the 28up + 8down-feISC signature were categorized as CMS4 (Fig. 5I), which is characterized by upregulation of epithelial-to-mesenchymal transition (EMT) gene signatures, TGFβ signaling, stromal infiltration, and poorer patient prognosis. In contrast, tumors displaying the opposite phenotype (28up-low+8down-high) were primarily ascribed to the more canonical Wnt and Myc-driven

**Fig. 5 CT-induced quiescent cells display a fetal intestinal stem cell signature that is partially dependent on p53. A** Expression correlation matrix from the 28up + 8down-feISC gene signature in Marisa database ($n = 566$). Size of circles and color intensity are proportional to Pearson correlation coefficient for each gene pair. **B** RT-qPCR analysis of normalized expression of selected 28up + 8down-feISC genes in untreated and treated PDO5 ($n = 2$–4 biologically independent experiments). **C** WB analysis of control and 5-FU + Iri-treated PDO5 cells collected at the indicated time points (from one out of three biologically independent experiments). **D–F** RT-qPCR analysis of normalized relative expression of indicated genes in control and CT-treated TP53 WT PDO66 **D**, TP53 mutant PDO4 **E** and PDO5 *TP53* KO #3 **F** ($n = 2$–4 biologically independent experiments). **G** WB analysis of PDO5 and PDO5 *TP53* KO untreated or treated with 5-FU + Iri. at the same concentration at the indicated time points (from one out of three biologically independent experiments). **H** WB analysis with indicated antibodies of CRC cells untreated or treated 72 h with 5-FU + Iri. (from one out of three biologically independent experiments). **I** Pie charts showing the molecular subtype distribution according to Guinney et al., in patients within the feISC signature groups as indicated. **J** Distribution of selected 28up-fetal-ISC genes in epithelial subtypes cell states 1–9 described by Lee et al.[36]. The *t*-SNE plots were obtained using the web-based tool URECA (User-friendly InteRface tool to Explore Cell Atlas). For all applicable figure panels, data are mean ± SD. Significance ($p$) was calculated with a two-sided Student's *t* test. *$p < 0.05$; **$p < 0.01$; ****$p < 0.0001$; n.s., no significant. CT, control; $IC_{20}$ and $IC_{30}$, 5-FU + Iri. treatment indicates 20 and 30% cell death, respectively. Source data are provided as a Source Data file.

---

CMS2 subtype. We studied whether the feISC signature of untreated tumors was expressed in the epithelial cancer cells or primarily contributed by the stromal component. Analysis of single-cell RNA-seq data from Lee and collaborators[36] demonstrated that feISC genes are expressed in the epithelial cancer cells, particularly in states 1, 5, and 6 from Lee and collaborators that are all associated with the secretory and migratory pathways (Fig. 5J).

These results identified a specific feISC signature that is induced by sublethal CT in a YAP1-dependent manner, but it is already present in the subset of CMS4 subtype (untreated) CRC tumors from Guinney and collaborators, and in the secretory and migratory epithelial states 1, 5, and 6 from Lee and collaborators.

**The YAP-dependent feISC signature is predictive of reduced disease-free survival in *TP53* WT tumors.** Our results indicated that feISC conversion was induced by YAP signaling, and nuclear YAP1 predicts a poor prognosis in untreated human CRC. Thus, we studied whether the 28up + 8down-feISC signature can be used as a prognosis tool in patients. The global 28up + 8down-feISC signature was sufficient to demarcate at least two subsets of patients in the Marisa (Fig. 6A and Supplementary Table S3), Jorissen, and TCGA data sets (Supplementary Table S3), within the group with the highest 28up and lowest 8down-feISC levels displaying the poorest disease-free-survival (Fig. 6B). A more detailed analysis of the Marisa data set demonstrated that this signature was significantly associated with tumor relapse in patients at stages II ($n = 264$) ($p = 0.041$) (Fig. 6C) and II + III ($n = 469$) ($p = 0.0033$) (Fig. 6D), and imposed a trend towards poor prognosis at stage IV ($n = 60$) (Fig. 6E).

The presence of functional p53 defines DNA repair efficiency and TIC of cells acquiring the feISC signature after damage (see Figs. 2A, B, 5F, G). Thus, we explored the possibility that *TP53* status determined the prognosis value of 28up + 8down-feISC signature in CRC patients. We observed increasing proportion of *TP53* mutated tumors according to tumor stage in the TCGA data set (Fig. S6A) as expected, however stratification of Marisa (Fig. 6F) and TCGA (Fig. S6B) patients according to *TP53* status did not have a prognosis value by itself. Importantly, *TP53* status reliably determines the prognosis value of 28up + 8down-feISC signature specific at stages II + III (Fig. 6G and Fig. S6C). Because 28up + 8down-feISC tumors are mainly included in the worst prognosis CMS4, we tested whether this feISC signature represents an independent prognosis factor inside this molecular subtype. Carrying the 28up + 8down-feISC signature increased the risk of relapse in patients within the CMS4 group (Fig. 6H). Interestingly, we observed a slight accumulation of *TP53* WT in tumors carrying the 28up + 8down-feISC signature in the subgroups of CMS4 from TCGA and Marisa cohorts (Fig. S6D).

Together with our results demonstrate the existence of a YAP1-dependent feISC signature that is induced after sublethal CT and favors cancer progression and metastasis. This signature predicts poor survival of CRC patients in the context of functional p53.

**Discussion**

We have here identified a YAP1-dependent feISC signature that can be induced by CT, associated with the acquisition of a PQL state, but that is already present in untreated tumors from several CRC cohorts. The molecular mechanisms inducing YAP1 activation by CT are not yet known, and we speculate that stromal populations present in the tumor, such as inflammatory cells, may induce upstream regulators of this signature (i.e., TGFβ signaling) thus leading to the acquisition of PQL traits before CT treatment. In agreement with this possibility, tumors carrying the 28up + 8down-feISC signature are primarily included in the CMS4 CRC subtype from Guinney and collaborators[35] characterized by stromal infiltration and TGFβ signaling. It was shown that TGFβ can promote YAP1 signaling by facilitating the degradation of the negative regulator of the pathway RASSF1A[37]. Moreover, identification of this feISC signature will allow the classification of patients with a higher probability of recurrence at diagnosis, which will benefit from more aggressive treatments or closer follow-up. Additionally, targeting the upstream signals imposing PQL/feISC acquisition pharmacologically (i.e. YAP1 or TGFβ inhibitors) or using combination treatments that effectively eradicate quiescent tumor cell populations (i.e. CT plus inhibitors of the NHEJ repair pathway) appear as interesting therapeutic options. Conversion of adult into fetal ISC had already been identified as part of the process of tissue regeneration after helminths infection[26] or in the Dextran Sulfate Sodium colitis model[29]. Thus, our results reinforce the concept that tumor development is partially mimicking the tissue regeneration process.

CT is the current treatment for advanced and metastatic colorectal tumors. However, in a percent of cases, tumor cells that escape from treatment (by efficient drug clearance, effective DNA repair, or reduced accessibility of the drugs) can acquire a dormant phenotype that could provide superior resistance to subsequent damaging-based treatments. In the present study we have shown that sublethal CT imposes a non-senescent and non-proliferating phenotype on cancer cells, in the absence of persistent DNA damage (we refer to this phenotype as PQL).

Importantly, PQL cells can efficiently escape from dormancy following in vivo transplantation, as it is shown in the intra-cecal xenograft experiments, in particular when cells migrate from the site of implantation. This is in agreement with their higher metastatic potential and is also in accordance with previous

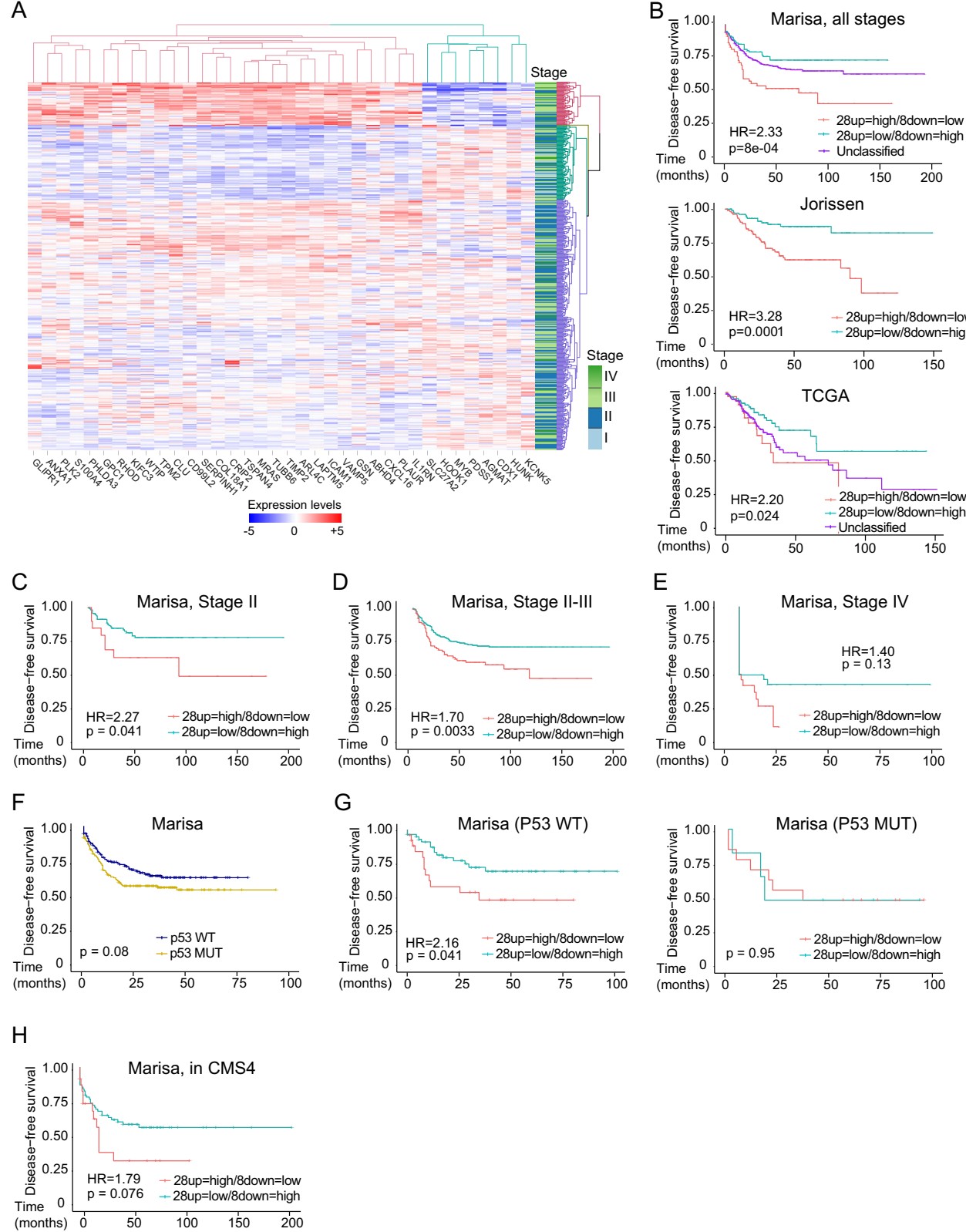

studies showing that dormant populations of primary human CRC cells retain tumor propagation potential[38] and cancer cells displaying reversible-quiescent states after CT exhibit increased tumorigenic potential[12]. Of note, whereas differences in tumor dissemination between untreated and CT-treated PDO5 in the intra-cecal transplantation experiments were highly significant, we only detected slight differences in the intracardiac assays.

These divergent results were likely indicating that intracardiac injection does not reflect all steps of the metastatic process such as EMT or invasive capacity. Supporting this possibility, EMT is one of the pathways that are enriched in the CT-treated PDOs (see Fig. 3A), and we have now confirmed that EMT genes are specifically upregulated in the quiescent cell population (see Fig. 3H).

**Fig. 6 Identification of a fetal ISC signature with prognosis value in CRC. A** Unsupervised hierarchical cluster analysis of patients according to 28up + 8down-feISC gene signature leading to the classification of patients into four subsets (colored in red, green, light blue, and purple). Tumor staging is indicated. **B** Kaplan–Meier representation of disease-free survival (DFS) over time for patients according to 28up + 8down-feISC signature selected according to **A** for Marisa (28up = high/8down = low $n = 66$, 28up = low/8down = high $n = 114$ and unclassified $n = 386$), Jorissen (28up = high/8down = low $n = 114$ and 28up = low/8down = high $n = 112$) and TCGA (28up = high/8down = low $n = 39$, 28up = low/8down = high $n = 96$ and unclassified $n = 194$) colorectal cancer databases. Patients were selected according to cluster analysis of the 28up + 8down-feISC signature. **C–E** Kaplan–Meier curves representing patients DFS classified according to cluster analysis of the 28up + 8down-feISC signature of stage II ($n = 149$) **C**, stage II and III ($n = 468$) **D**, and stage IV ($n = 60$) **E** patients from Marisa database. **F** Kaplan–Meier representation of patients DFS classified according to TP53 in the Marisa data set (TP53 WT $n = 161$ and TP53 MUT $n = 190$). **G** Kaplan–Meier representation of DFS of patients from the Marisa data set classified according to cluster analysis of the 28up + 8down-feISC signature in TP53 WT ($n = 85$) and TP53 mutant ($n = 20$) groups. **H** Kaplan–Meier representation of DFS in CMS4[35] patients from Marisa cohort classified based on 28up + 8down-feISC signature ($n = 91$). Data in **A** show normalized, centered, and scaled Illumina probe set intensities on a log2 scale. The stage lane represents the subtype corresponding to each patient. We used Cox proportional hazards models for statistical Kaplan–Meier analysis and log-rank two-sided $p$ value (see Supplementary Table S3). HR, hazard ratio.

Our transcriptomic studies revealed that acquisition of PQL phenotype rely on p53 and p21 signaling. It has long been established that the key regulatory proteins that mediate cell cycle block include p53, p21 and p16, among others (reviewed in ref.[39]). Notably, cancer cells carrying dysfunctional p53 show the partial conversion to fetal phenotype but fail to acquire a functional PQL phenotype, as they lose TIC associated with massive accumulation of DNA damage, when treated at doses that reduce cell growth ~20–30 %. Although it is still possible that p53 mutated cells can acquire a PQL phenotype at specific CT doses, we found that p53 deletion by itself led to the accumulation of DNA damage and imposed defective clonogenic activity after exposure to comparable levels of 5-FU + Iri.

Our findings, which are in agreement with the recent demonstration that tumor cells that persist after prolonged CT acquires an embryonic-like and quiescent state that facilitates therapeutic resistance[14] will also help to clarify the functional contribution of YAP1 as a driver of fetal conversion and driver or suppressor of metastasis (reviewed in ref.[40]). We propose that functional p53 through p21 allows cells to recover from DNA damage under specific conditions, whereas p53 mutant cells continue proliferating leading to irreparable damage and apoptotic death. This possibility would reconcile our observations with a previous publication indicating that fetal reprograming of intestinal cancer cells induced by YAP1 led to tumor and metastasis suppression in the Apc−/−; KrasG12D; p53−/− murine cancer model[16]. Specifically, our results indicate that fetal ISC conversion is not induced by p53 transcriptionally, but p53 protects fetal-converted cells from massive DNA damage and loss of TIC and metastatic potential after sublethal CT. This cellular response represents, in fact, a double-edged sword since it can impose specific outcomes depending on the TP53 status of cancer cells. In this sense, 5-FU treatment induces cell dormancy and epithelial-to mesenchyme transition in lung cancer cells, associated with p53 accumulation[41]. Further experiments genetically deleting YAP1 in TP53 WT cells and preclinical assays using YAP1 inhibitors are required to demonstrate the possibility of designing specific protocols to treat fetal-converted tumors.

Our findings linking fetal ISC conversion with poor CRC prognosis are partially opposite to the idea that adult ISC signatures are indicative of tumor malignancy[3]. Nevertheless, it has been recently demonstrated that Lgr5 and other adult ISC markers are temporarily lost from cells seeding metastases, and subsequently recovered (due to cellular plasticity) to allow metastasis establishment[42]. Moreover, most data indicating the requirement for adult ISCs in metastasis seeding were obtained on TP53 -deficient tumor cells[16,43]. Recently, it was found that Lgr5+ cells can differentially contribute to tumor progression and authors showed higher Lgr5+ frequency in mutant TP53[44], thus opening the possibility that Lgr5+ ISCs dependence in cancer is linked to TP53 status. Further studies are required to clarify these discrepancies. Independently on the mechanisms underlying fetal ISCs conversion, we have here identified a restricted genetic signature that is present in a subset of tumors that are mostly included in the CMS4 but clearly different from that of adult ISCs.

From a clinical perspective, uncovering genetic signatures that are predictive of recurrence in a group of patients with uncertain projections (stages II and III) will represent a powerful tool for diagnosis refinement. In this direction, we are currently setting up the protocols for early detection of PQL/feISC cells in stages II–III tumors at diagnosis. As mentioned, anticipating the presence of this adverse phenotype in tumors would allow exposing selected groups of patients to alternative therapeutic procedures that could be refined with the discovery of the mechanisms imposing fetal SC conversion in cancer. Moreover, our data alert about potential negative effects of neoadjuvant treatment and indicate the possibility of using nuclear YAP1 or specific fetal ISC markers to identify patients that would benefit from future YAP1-based therapeutic strategies.

## Methods

**Study design.** The goal of this study was to determine the impact of sublethal CT in colorectal cancer cells. A study of sublethal doses of CT was performed in several PDOs and human cell lines. We identified genetic signatures induced by CT that were evaluated in public colorectal tumor databases for their prognosis value. Numbers of experiments, biological replicates, and sample sizes are outlined in the figure legends.

**Cell lines.** CRC cell lines HCT116 and Ls174T (KRAS mutated and TP53 WT), SW480 (KRAS and TP53 mutated), and HT29 (BRAF and TP53 mutated) were obtained from the American Type Culture Collection (ATCC, USA). All cells were grown in Dulbecco's modified Eagle's medium (Invitrogen) plus 10% fetal bovine serum (Biological Industries) and were maintained in a 5% $CO_2$ incubator at 37 °C. 5-FU + Iri. concentrations that reduced 30% of each cell growth were as follows: HCT116, 0.01 μg/mL 5-FU and 0.004 μg/mL Iri.; Ls174T, 0.025 μg/mL 5-FU and 0.01 μg/mL Iri.; SW480, 0.28 μg/mL 5-FU and 0.11 μg/mL Iri.; HT29, 0.33 μg/mL 5-FU and 0.13 μg/mL Iri.

**Human colorectal tumors.** Formalin-fixed, paraffin-embedded tissue blocks of gastrointestinal tumor samples, from patients at diagnosis after neoadjuvant therapy at the time of surgery, were obtained from Parc de Salut Mar Biobank (MARBiobank, Barcelona). Samples were retrieved under informed consent and approval of the Clinical Research Ethics Committee-Parc de Salut Mar (CEIC-PSMAR) according to Spanish ethical regulations and the guidelines of the Declaration of Helsinki. Patient identity remained anonymous in the context of this study. Patient data was collected and treatment regimens were standard and adjusted to patient and tumor characteristics (see Supplementary Tables S2 and S6). IHC analyses were performed as described below.

**Animal studies.** For tumor-initiating/metastasis assays we followed two approaches: 1) Intracardiac injection of 40,000 control, IC20 or IC30-treated PDO5 cells carrying a luciferase reporter to NSG mice in two independent experiments (strain: ANB//NOD.Cg-Prkdcscid Il2rgtm1Wjl/SzJ; ten-week-old males). To determine

tumor load, animals were anesthetized and injected with 100 µl of substrate D-luciferin at 15 mg/ml intraorbital. Bioluminescent images were taken at day 0 and every week in the IVIS Lumina III In Vivo Imaging System (PerkinElmer) with 2 min exposure. Metastasis initiation differences were determined by using the two-way ANOVA test. Quantification was done using Living Image® software (PerkinElmer). 2) Equivalent amounts of disaggregated PDO cells, previously treated as indicated, were implanted as orthoxenografts athymic nude mice (strain: Hsd:Athymic Nude-Foxn1nu; 5–7-week-old males). Tumor growth was determined by palpation, and animals were sacrificed when controls developed tumors of around 2 cm in diameter. Procedures involving living animals were conducted under pathogen-free conditions and according to guidelines from the Animal Care Committee at the Generalitat de Catalunya. The Committee for Animal Experimentation at the Institute of Biomedical Research of Bellvitge (Barcelona) approved these studies.

**PDO and culture conditions**. Human colorectal tumors were obtained from Parc de Salut MAR Biobank (MARbiobank) and IdiPAZ Biobank, integrated into the Spanish Hospital Biobanks Network (RetBioH; www.redbiobancos.es). Written informed consent was obtained from all participants and protocols were approved by Hospital del Mar' Ethics Committee (approval number 2019/8595/I), the Spanish regulations, and the Helsinki declaration's Guide. For PDOs generation, primary or xenografted human colorectal tumors were disaggregated in 1.5 mg/mL collagenase II and 20 µg/mL hyaluronidase after 40 min of incubation at 37 °C, filtered in 100 µm cell strainer, and seeded in 50 µl Matrigel in 24-wells plates. After polymerization, 450 µL of complete medium was added (DMEM/F12 plus penicillin (100 U/mL) and streptomycin (100 µg/mL), 100 µg/mL Primocin, 1× N2 and B27, 10 mM Nicotinamide; 1.25 mM N-Acetyl-L-cysteine, 100 ng/mL Noggin and 100 ng/mL R-spondin-1, 10 µM Y-27632, 10 nM PGE2, 3 µM SB202190, 0.5 µM A-8301, 50 ng/mL EGF and 10 nM Gastrin I). Tumor spheres were collected and digested with an adequate amount of trypsin to single cells and re-plated in culture. Cultures were maintained at 37 °C, 5% $CO_2$ and medium changed every week. PDOs were expanded by serial passaging and kept frozen in liquid Nitrogen for being used in subsequent experiments. Mutations identified in the PDOs are listed in Supplementary Table S1.

**PDO viability assays**. 600 single PDO cells were plated in 96-well plates in 10 µL Matrigel with 100 µL of complete medium. After 6 days in culture, growing PDOs were treated with combinations of 5-FU + Iri. for 72 h at the concentrations that reduce 20 and 30% of the cell growth ($IC_{20}$ and $IC_{30}$, respectively), which are specific for each PDO as described in Supplementary Table S1. After 72 h of treatment, we changed to a fresh medium and measured the cell viability after 3 days, 1 week, and 2 weeks using the CellTiter-Glo 3D Cell Viability Assay following manufacturer's instructions in an Orion II multiplate luminometer. Images were obtained with an Olympus BX61 microscope at the indicated time points and the diameter of at least 70 tumoroids per condition was determined using Adobe Photoshop. For dose–response curves, PDOs were plated in 96-well plates in Matrigel and after 6 days in culture were treated with combinations of 5-FU and Irinotecan. Following 72 h of treatment, we changed to fresh medium and treated with increasing concentrations of either 5-FU + Iri., dasatinib, verteporfin or combinations for 72 h at the indicated concentrations. Cell viability was determined as described above.

**TIC assay**. For TIC assay in vitro, 300 or 600 single PDO cells were plated in 96-well plates in 10 µL Matrigel. After 11 days in culture, the number of PDOs in each well was counted, photographs were taken for PDO diameter determination and cell viability was measured.

**PDOs infection**. hFLiG plasmid was used for in vivo detection of metastasis, H2BeGFP plasmid was used for flow cytometry experiments and lentiCRISPR v2 was used for knock-out experiments. Three sgRNA against TP53 and YAP1 gene were designed using Benchling (Supplementary Table S5). Lentiviral production was performed by transfecting HEK293T cells the lentiviral vectors and the plasmid of interest. One day after transfection, the medium was changed, and viral particles were collected 24 h later and then concentrated using Lenti-X Concentrator. PDOs were infected by resuspending single cells in concentrated virus diluted in a complete medium, centrifuged for 1 h at 650 rcf, and incubated for 5 h at 37 °C. Cells were then washed in a complete culture medium and seeded as described above.

**Immunohistochemical staining**. Paraffin blocks were obtained from tissues and PDOs, the previous fixation in 4% formaldehyde overnight at room temperature. Paraffin-embedded sections of 4 µm, for tissues, and 2.5 µm, for PDOs, were de-paraffinized, rehydrated and endogenous peroxidase activity was quenched (20 min, 1.5% $H_2O_2$). EDTA or citrate-based antigen retrieval was used depending on the primary antibody used. All primary antibodies were diluted in PBS containing 0.05% BSA, incubated overnight at 4 °C, and developed with the Envision+ System HRP Labeled Polymer anti-Rabbit or anti-Mouse and 3,3'-diaminobenzidine (DAB). Samples were mounted in DPX and images were obtained with an Olympus BX61 microscope.

**Immunofluorescence analysis**. For tissues and PDOs, the same protocol as IHC was followed. However, the samples were developed with Tyramide Signal Amplification System (TSA) and mounted in DAPI Fluoromount-G. Images were taken in an SP5 upright confocal microscope (Leica).

**Hematoxylin and eosin staining**. Previously de-paraffinized sections were incubated with hematoxylin 30 s, tap water 5 min, 80% ethanol 0.15% HCl 30 s, water 30 s, 30% ammonia water (NH3(aq)) 30 s, water 30 s, 96% ethanol 5 min, eosin 3 s, and absolute ethanol 1 min. Samples were dehydrated, and mounted in DPX, and images were obtained with an Olympus BX61 microscope.

**FISH**. Fluorescent in situ hybridization (FISH) analyses from untreated and $IC_{30}$-treated PDOs were performed using commercial probes (Abbott Molecular Inc, Des Plaines, IL, USA), one including the centromeric alfa-satellite region specific for chromosome 8, and a second one containing locus-specific probes from the long arm of chromosome 13 and 21. In brief, we performed a cytospin to concentrate nuclei in the FISH slide. Slides were pre-treated with pepsin for 5 min at 37 °C. Samples and probes were co-denatured at 80 °C for five minutes and hybridized overnight at 37 °C in a hot plate (Hybrite chamber, Abbot Molecular Inc.). Post-hybridization washes were performed at 73 °C in 2× sodium salt citrate buffer (SSC) and at room temperature in 2× SSC, 0.1% NP-40 solution. Samples were counterstained with 4,6-diamino-2-phenilindole (DAPI)(Abbott Molecular Inc, Des Plaines, IL, USA). Results were analyzed in a fluorescence microscope (Olympus, BX51) using the Cytovision software (Applied Imaging, Santa Clara, CA). A minimum of 50 nuclei per case was analyzed.

**Comet assay**. Comet assays were performed using Comet Assay Kit following manufacturer's instructions. Pictures were taken using a Nikon Eclipse Ni-E epi-fluorescence microscope and tail moment was calculated using the OPENCOMET plugin for Fiji.

**Annexin V binding assay**. Annexin V binding was determined by flow cytometry using the standard Annexin V Apoptosis Detection Kit APC. Single cells of treated PDOs with indicated combinations of 5-FU + Iri. were obtained and stained according to the manufacturer's instructions, with Propidium Iodide staining for the DNA content. The cells were analyzed in the Fortessa analyzer.

**Cell senescence assays**. Cell senescence was identified by the presence of SA-β-galactosidase activity using two different approaches. On one hand, staining for SA-β-galactosidase activity in cultured cells was carried out using the Senescence β-Galactosidase Staining Kit. Briefly, PDOs were seeded in 24-well plates (3000 cells per well). After 6 days, PDOs were treated with combinations of 5-FU + Iri. for 72 h and were subsequently stained with the β-Galactosidase Staining Solution for 2 h, according to the manufacturer's instructions. Sections embedded in paraffin were counterstained with Fast Red for nuclei visualization. Images were obtained with an Olympus BX61 microscope. On the other hand, SA-β-galactosidase activity was addressed by flow cytometry using the Cell Event Senescence Green Flow Cytometry Assay Kit following the manufacturer's instructions, and analyzed in the LSR II analyzer.

**Cell cycle analysis**. Cell cycle was determined by flow cytometry using the standard APC BrdU Flow Kit. Briefly, treated PDOs with combinations of 5-FU + Iri., as indicated, were stained with bromodeoxyuridine (BrdU) for 24 h. Single cells were obtained and processed according to the manufacturer's instructions, with DAPI staining for the DNA content. The cells were analyzed in the LSR II analyzer.

**Cell lysis and Western Blot**. Treated PDOs were lysed for 20 min on ice in 300 µL of PBS plus 0.5% Triton X-100, 1 mM EDTA, 100 mM NA-orthovanadate, 0.2 mM phenyl-methylsulfonyl fluoride, and complete protease and phosphatase inhibitor cocktails. Lysates were analyzed by western blotting using standard SDS–polyacrylamide gel electrophoresis (SDS-PAGE) techniques. In brief, protein samples were boiled in Laemmli buffer, run in polyacrylamide gels, and transferred onto polyvinylidene difluoride (PVDF) membranes. The membranes were incubated with the appropriate primary antibodies overnight at 4 °C, washed, and incubated with specific secondary horseradish peroxidase–linked antibodies. Peroxidase activity was visualized using the enhanced chemiluminescence reagent and autoradiography films.

**RT-qPCR analysis**. Total RNA from treated PDOs was extracted with the RNeasy Micro Kit, and cDNA was produced with the RT-First Strand cDNA Synthesis Kit. RT-qPCR was performed in LightCycler 480 system using SYBR Green I Master Kit. Samples were normalized relative to the housekeeping genes TBP and/or HPRT1. Primers used for RT-qPCR are listed in Supplementary Table S5.

**RNA-sequencing and analysis**. Total RNA from untreated and treated PDOs was extracted using RNeasy Micro Kit. The RNA concentration and integrity were determined using Agilent Bioanalyzer [Agilent Technologies]. Libraries were

prepared at the Genomics Unit of PRBB (Barcelona, Spain) using standard protocols, and cDNA was sequenced using Illumina HiSeq platform, obtaining ~45–64 million 50-bp paired-end reads per sample. Adapter sequences were trimmed with Trim Galore. Sequences were filtered by quality ($Q > 30$) and length (>20 bp). Filtered reads were mapped against the latest release of the human reference genome (hg38) using default parameters of TopHat (v.2.1.1)[1] and expressed transcripts were then assembled. High-quality alignments were fed to HTSeq (v.0.9.1)[2] to estimate the normalized counts of each expressed gene.

Differentially expressed genes between conditions (considering $IC_{20}$-, $IC_{30}$-treated PDO5, and $IC_{30}$-treated PDO66 as treated conditions and untreated samples from PD05 and PDO66 as control conditions) from RNA-seq data were explored using DESeq2 R package (v.1.30.1)[45] and the removeBatchEffect from the limma package (v.3.46.0) was used to correct technical batch effects. Adjusted $P$ values for multiple comparisons were calculated applying the Benjamini-Hochberg correction (False Discovery Rate) (see Supplementary Data 1). Plots were done in R. Expression heatmaps were generated using the heatmaply and pheatmap packages in R[46]. GSEA was performed with described gene sets using gene set permutations ($n = 1000$) for the assessment of significance and signal-to-noise metric for ranking genes. RNA-sequencing data for PDO5 and PDO66 are deposited at the GEO database with accession number GSE155354.

**ChIP-sequencing and analysis**. Formaldehyde cross-linked cell extracts of IC20-treated PDO5 were sonicated, and chromatin fractions were incubated for 16 h with anti-p53 [abcam ab 1101] antibody in RIPA buffer and then precipitated with protein A/G-sepharose [GE Healthcare, Refs. 17-0618-01 and 17-0780-01]. Crosslinkage was reversed, and 6–10 ng of precipitated chromatin was directly sequenced in the genomics facility of Parc de Recerca Biomèdica de Barcelona (PRBB) using Illumina® HiSeq platform. Raw single-end 50-bp sequences were filtered by quality ($Q > 30$) and length (length > 20 bp) with Trim Galore[4]. Filtered sequences were aligned against the reference genome (hg38) with Bowtie2[5]. MACS2 software[6] was run first for each replicate using unique alignments ($q$ value < 0.1). Peak annotation was performed with ChIPseeker package[7] and peak visualization was done with Integrative Genomics Viewer (IGV). ChIP-sequencing data are deposited at the GEO database with accession number GSE164161.

**ChIP-qPCR**. Untreated and IC20-treated PDOs were subjected to ChIP following standard procedures. Briefly, PDO cells were extracted with formaldehyde cross-linked for 10 min at room temperature and lysed for 20 min on ice with 500 μL of $H_2O$ plus 10 mM Tris-HCl pH 8.0, 0.25% Triton X-100, 10 mM EDTA, 0.5 mM EGTA, 20 mM β-glycerol-phosphate, 100 mM NA-orthovanadate, 10 mM NaButyrate and complete protease inhibitor cocktail. The supernatants were sonicated, centrifuged at 13,000 rpm for 15 min, and supernatants were incubated overnight with anti-p53 antibody in RIPA buffer. Precipitates were captured with 35 mL of protein A-Sepharose, extensively washed, and analyzed by ChIP-qPCR. The primers used are listed in Supplementary Table S5. Inputs were used to normalize the ChIP-qPCR and samples were compared to control IgGs.

**Description of the patient gene expression data sets**. Transcriptomic and available clinical data sets from colorectal cancer were downloaded from the open-access resource CANCERTOOL. For CRC we used the Marisa (GSE39582) data set, which includes expression and clinical data for 566 patients with CRC and 19 non-tumoral colorectal mucosa, the Jorissen (GSE14333) data set and the TCGA data set with expression and clinical data of 226 and 329 CRC patients, respectively.

**Association of the signatures with clinical outcome**. Association of the expression of the signature with relapse was assessed in the cancer transcriptomic data sets using Kaplan–Meier estimates and Cox proportional hazard models. A standard log-rank test was applied to assess significance between groups. This test was selected because it assumes the randomness of the possible censorship. All the survival analyses and graphs were performed with R using the survival (v.3.2-3) and survimer (v.0.4.8) packages and a p value<0.05 was considered statistically significant (see Supplementary Table S3).

**Signature definition**. To generate the feISC signatures, we selected genes with log2 Fold Change (FC) TreatedvsControl > 0 and FetalvsAdult[30] >0 in the case of the 28up-feISC and $\log^2$FC TreatedvsControl < 0 and FetalvsAdult < 0 in the case of the 8down-feISC. Next, we used the Marisa data set to perform expression correlation matrices for the selected expression gene pairs using the corrplot package (v.0.84). Correlations were considered statistically significant when the Pearson correlation coefficient corresponded to a p value below 0.05. Clusters of genes were selected when the absolute value for the Pearson correlation coefficient was above 0.1.

**Quantification and Statistical analysis**. Statistical parameters, including the number of events quantified, standard deviation, and statistical significance, are reported in the figures and in the figure legends. Statistical analysis has been performed using GraphPad Prism 6 software, and P < 0.05 is considered significant. Two-sided Student's $t$ test was used to compare differences between two groups.

Each experiment shown in the manuscript has been repeated at least twice. Combinations of 5-FU + Iri. treatment has been checked for an appropriate $IC_{20}$ and $IC_{30}$ effect in every experiment, by cell viability assay. Bioinformatic analyses were performed as indicated above.

**Reporting summary**. Further information on research design is available in the Nature Research Reporting Summary linked to this article.

## Data availability

RNA sequencing and ChIP-sequencing data have been deposited in NCBI's Gene Expression and are accessible through GEO Series accession no. GSE155354 and no. GSE164161 respectively. Public data sets used in this study are also accessible through GEO Series accession no. GSE39582 for Marisa data set and GSE14333 for Jorissen data set; and for TCGA data set through the TCGA portal [http://www.tcgaportal.org/]. In this study, data from these data sets have been obtained using CANCERTOOL platform [http://genomics.cicbiogune.es/CANCERTOOL/]. Data from the experiments are provided in the Supplementary Information. Source data are provided with this paper.

## Code availability

No new codes were used in this study.

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

## Acknowledgements

We want to thank the Bigas' and Espinosa's lab members for constructive discussions and suggestions and technical support. We thank our patients for their generosity and to MARbiobank and the IdiPAZ Biobank integrated with the Spanish Hospital Biobanks Network (RetBioH; www.redbiobancos.es). This work was funded by Instituto de Salud Carlos III (ISCIII) and co-funded by the European Union (PI16/00437 and PI19/00013), Generalitat de Catalunya 2017SGR135, PID2019-104867RB-I00/AEI/10.13039/501100011033 funded by the Agencia Estatal de Investigación and the "Xarxa de Bancs de tumors sponsored by Pla Director d'Oncologia de Catalunya (XBTC), Fundación HNA to A.V. research team in the development of orthoxenografts/PDOX., Instituto de Salud Carlos III-Fondo Europeo de Desarrollo Regional (CIBERONC; CB16/12/00244, CB16/12/00241 and CB16/12/00273). IdiPAZ Biobank is supported by Instituto de Salud Carlos III, Spanish Health Ministry (Grant RD09/0076/00073), and Farmaindustria through the Cooperation Program in Clinical and Translational Research of the Community of Madrid. LS is supported by AGAUR (2018 FI_B 00088/2020 FI_B2 00150). D.A-V was funded by FI20/00130 and Y.G. and T.L-J by a contract from CIBERONC (ISCIII-Feder) and Asociación Española Contra el Cancer (AECC). T.C-T was funded by the Instituto de Salud Carlos III-FSE (MS17/ 00037; PI18/00014). A.V was supported by grant PI19/01320 funded by Instituto de Salud Carlos III (ISCIII) and co-funded by the European Union.

## Author contributions

Conceptualization: A.B., L.E., biochemical assays, in vitro and in vivo experiments: A.Vi., M.G., M.S., R.G.-R., M.M.-I., I.S., D.A.-V., J.A.-M., J.G., C.R. Experiments, investigation and evaluation of results: A.Ba., L.S., T.L.-J., A.Ve., T.C.-T., A.M., L.E. Big data analysis and statistical analysis: T.L.-J., Y.G., E.L.-A., F.T. Clinicopathological characterization of human tumors: M.G., R.S., J.L., M.I., C.M. Clinical advice: M.G., B.B., R.S., C.M. Writing–original drat: L.E., A.Bi. Writing–review & editing: all authors

## Competing interests

Laura Solé, Teresa Lobo-Jarne, Marta Guix, Beatriz Bellosillo, Mar Iglesias, Anna Bigas, and Lluís Espinosa have a pending patent application entitled Genomic predictor of outcome in cancer with number PCT/EP2022/058503, related to the fetal signature as a prognostic biomarker in colorectal cancer. The authors have no additional financial interests and the remaining authors declare no competing interests.
