## [Peer review file · Nature Communications]

REVIEWER COMMENTS

Reviewer #1 (Remarks to the Author):

Fetal conversion in p53 wildtype colorectal cancer cells imposes higher metastatic activity and poor prognosis

Sole and co-authors.

The authors seek to demonstrate that colorectal cancers which retain wildtype p53 acquire a quiescence-like phenotype following administration of a sub-lethal dose of a clinically relevant genotoxic chemotherapeutic. They go on to demonstrate that this approach also results in induction of a fetal-like intestinal stem cell associated transcriptional program, which is dependent upon YAP1. The authors go on to show that nuclear YAP1 is associated with poor outcomes in human colorectal cancers which retain wildtype p53.

Comments to the authors –

I have a number of concerns related to this manuscript at present –

In the initial section of the manuscript “Low-dose CT treatment induces a non-senescent quiescent-like phenotype to TP53 WT cancer cell in the absence of sustained DNA damage”

1) The authors have gone to significant efforts to determine sub-lethal doses of 5-FU and Irinotecan (henceforth CT) in their collection of colorectal cancer PDOs. They go on to show that administration of these sublethal CT doses, either IC20 or IC30 for 72 hours results in a robust growth arrest which persists even following washout for 2 weeks.

a. At this point, the authors characterise in depth the growth arrested phenotype, albeit only in PDO5 (p53 WT), despite the growth arrest phenotype appearing more robust in p53 mutant lines PDO4, PDO10, PDO11 and PDO15 (although they do not mention PDO4 at this point).

b. The authors go on to show that IC20 and IC30 doses of CT induce substantial DNA damage in p53 mutant lines PDO4 and PDO8, but not in the p53 wildtype line PDO5. Leading to the statement that the presence of p53 permits acquisition of the PQL phenotype in the absence of DNA damage.

These observations raise a number of questions –

i) Is the phenotype ascribed to PDO5 ie quiescence in the absence of DNA damage shared by other p53 wildtype lines, or is it unique to this line? The authors mention PDO66 in later experiments (but share very little data related to this line) – does it exhibit the same phenotype? Are other p53 wildtype lines available to test?

ii) When looking at the DNA damage phenotype associated with p53 mutation/loss, does the PDO8 line exhibit the same growth-arrest phenotype as all other PDOs mentioned following administration of

sublethal CT? Why do the authors demonstrate growth arrest in PDOs 4, 5, 10, 11 and 15, but assess DNA damage phenotype in PDOs 4, 5 and 8?

iii) P53 loss or mutation is associated with tolerance of increased or sustained DNA damage, and is associated resistance to genotoxic CT regimens. This is reflected by the absolute dose of CT administered as IC20 + IC30 for the individual lines studied, where for example the IC20 values for PDO4 were 1.25ug/ml 5-FU/0.5ug/ml Irinotecan, while for PDO5 these were 0.14ug/ml 5-FU/0.06ug/ml irinotecan (roughly 10x lower for each). It is conceivable that observed DNA damage phenotype observed in the p53 mutant lines arises solely because they are able to survive significantly higher levels of DNA damage driven by higher doses of CT, while the p53 WT lines are not. The phenotype could be that the IC20 of CT in PDO5 is at a level below that required to elicit significant DNA damage, but high enough to drive quiescence. In order to demonstrate that the adoption of a quiescent state in the absence of increased/sustained DNA damage is a unique feature of p53 WT cells, the authors should assess both DNA damage phenotypes and quiescence of p53 mutant/deficient lines at lower doses of CT (eg the IC20 of PDO5).

iv) The question raised in iii) above could be partially addressed in Figure 1K, where the authors demonstrate induction of DNA damage reflected by γ H2A.X induction following P53 deletion in PDO5, but this lacks clarity. It is notable that the sublethal CT treatment here does not increase γ H2A.X above the already high basal level, in contrast to the p53 mutant lines discussed previously (e.g PDO4, PDO8; Fig S1H). Is this dose given to the PDO5 p53 KO line at the IC20 generated for the parental PDO5 line, or is it a newly generated IC20 for this specific P53 KO line? I think this points to the suggestion in point iii) above that the IC20 dose for PDO5 does not induce a strong DNA damage response under any circumstances. As a minor point, why in the control lane (PDO5, untreated) for this panel is P53 detectable, in contrast eg to Figure 4a?

This section does not convince me that the DNA damage phenotype in the p53 mutant lines is linked to the PQL phenotype (or indeed the absence of DNA damage in the p53 wildtype line), and indeed therefore that the presence of wildtype p53 is the key factor. This has an impact for the remainder of the study – for example the TIC experiments in the next section.

In the second section of the manuscript, entitled “PQL cancer cells display increased in vitro and in vivo tumor initiation capacity” -

1) I have some concern about the description of the use of PDO66 in this section of the manuscript (and elsewhere), primarily that the authors cite data related to PDO66, but do not include any data in the either the primary figure or supplementary to support their findings (related to figure 2A). Moreover, we do not have any data from the earlier section to suggest that PDO66 exhibits the same phenotype in terms of quiescence (the “PQL” phenotype), and absence of DNA damage as described for PDO5. This is a key point.

2) The intracardiac injection of PDO5 following administration of CT suggests an interesting phenotype, suggesting that sublethal CT does induce a metastatic phenotype/survival advantage to PDO5, although unfortunately appears underpowered. It would also be useful to know whether p53 mutant/deficient lines behave in the same way.

3) The section related to caecal transplantation is confusing – the text states that “IC20 and IC30-treated derived tumours significantly smaller than those arising from untreated controls (Figure 2F)” – but Figure 2F relates to number rather than size. Should this be Figure 2G, a measure of inoculation site vs peritoneal dissemination? In figure 2G, it is not clear what the tumour weight measurement actually refers to – is it the average cumulative weight of all IP implants found within each mouse?

As previously, while the phenotype observed under the conditions tested are absolutely correct, it is not fair to suggest that they are representative of P53 wildtype colorectal cancers as a whole, and may equally be a product of experimental approach taken.

Section entitled “CT-induced PQL cells display fetal intestinal stem cells (feISC) characteristics” –

The induction of fetal/regenerative stem cell like signatures following sublethal CT in this setting is very interesting and may be indicative of regeneration/recovery following loss of the *Igr5* canonical/homeostatic stem cell population. It is great to note that the observations related to PDO5 seem to be supported by data generated from PDO66, although again this must be fully presented for the reader rather than stated in the text – specifically the shift towards fetal phenotype in PDO66 must be shown rather than “not depicted”. If the association of this phenotype and earlier sections are to stand up, it is important to demonstrate here that this fetal regenerative transcriptional signature is not induced in p53 mutant or deficient line eg PDO4, 8 or the PDO5 p53 KO line (I note that this has been attempted in later sections). It is also important here to test both at the IC20/30 of the p53 deficient/mutant lines, and at the dose given to PDO5.

Section entitled “Acquisition of FeISC by CT treatments is linked to and dependent upon YAP1 activation” –

Putting aside my concerns around the use of a single organoid line, and the link between p53 status and the observed phenotype, this section has a number of key, and very important observations – critically the induction of nuclear YAP1 and fetal-associated markers in human CRC samples post neoadjuvant CT, when compared to paired pre-treatment biopsies. Although the link to “sublethal CT” in line with the in vitro data generated in the manuscript is tenuous.

Section entitled – “A restricted YAP1-dependent fetal signature shows co-ordinate expression in human CRC associated with higher p21 levels”

This section really nicely describes the generation of a canonical signature associated with the feISC population, and subsequent use across multiple human cohorts. Again, where a study has been carried, I would ask that the authors include the data, in this case in reference to the association between p21 in the Jorissen and TCGA datasets. The correlation of the feISC signature generated here with the CM4 (and anti-correlation with CMS2) from the Guinney et al 2015 paper is very interesting, and demonstrates really nicely that the fetal program may be associated with more aggressive disease (and a poorer prognosis). Given that around 50% of patients in CMS2 and CMS4 groups from the Guinney et

al dataset exhibit p53 mutation, it would be very interesting to know how closely the 28up+8down signature aligns to p53 status in these groups.

Section entitled – “The Yap-dependent feISC signature is predictive of reduced disease-free survival in TP53 WT tumors”

As with the previous section, this data is really nice – it demonstrates clearly the prognostic value of YAP1 dependent feISC signatures in human CRC. While the authors take encouragement from the link between TP53 status, the 28up-8down signature and prognosis in stage II+III patients, it is unusual to focus on this, when the data indicates that the 28up-8down signature is a good prognosticator in unselected patient populations in all datasets studied – ie it is not selective for p53 status.

Reviewer #2 (Remarks to the Author):

In this study Sole and colleagues investigate the role of Yap dependent fetal conversion in mediating tumour cell therapy resistance. They find that in P53 wild-type colorectal tumour organoids treatment with a sub-lethal dose of chemotherapy induces a quiescent, fetal-like state on tumour cells. These cells have high levels of Yap activity and show some evidence of increased metastatic potential. They further show that tumour cells with these properties are found in patient samples and are predictive of poor disease outcome. The concepts described here are interesting and could potential indicate a mechanism to target therapy resistant colorectal cancer cells via targeting the Yap pathway. Although the role of Yap in fetal conversion and the induction of quiescent like phenotypes following therapy have been previously described the linking of these states appears novel, timely and solidifies the evidence of the importance of these processes. However, a number of the key conclusions of the manuscript require further strengthening prior to publication, in particular the role of these cells in metastasis and the potential role of Yap1 in this process.

1) One of the key findings of the paper is that sub-lethal chemotherapy doses tumour cells have increased tumorigenicity in vivo and increased metastatic potential (Figure 2). However, the first of these experiments did not show a statistically significant difference (2C-E) negating the importance of this finding. In addition, it appears as though bulk PDO cells have been used for this experiment rather than the quiescent population in question. Thus, it is not clear which cells following treatment have metastatic capacity. The authors should repeat these experiments with quiescent cells using the H2B-GFP strategy employed for other experiments.

2) In addition to this, the role of Yap1 in imparting the fetal-like transcriptional signature is investigated but functional analysis of how this relates to tumour phenotypes is lacking. The authors should utilise

Yap1 KO PDO lines to determine whether this affects the tumour initiating and/or metastatic capacity of sub-lethal dosed PDO cells.

3) The relative role of the quiescent vs non quiescent cells in PDOs following sub-lethal chemotherapy is not completely clear. The authors carry out an in vitro experiment demonstrating similar colony forming capacity but some additional characterisation would be useful. Are there differences in Yap activity / marker gene expression for example? Are there any phenotypic differences in culture other than proliferation? Have they preferentially induced an EMT like programme?

4) The phenotypes observed appear to be specific for P53 wild-type tumours but the mechanism isn't explored. One possible explanation raised by the authors is that P53 allows survival of the Yap activated cells. The authors could investigate this possibility by analysing the P53 KO PDO5 model over a time course for Yap activity and apoptosis induction.

Minor points:

1) There is a lack of quantification throughout the manuscript. This is particularly apparent in Figure 1 where the key findings of the quiescent cell population are described. Quantification and statistical analysis of 1B, 1C, 1E, 1F and 1H should be included.

2) Same as above for Figure 4B – quantification of Yap1 nuclear localisation % should be included.

Reviewer #3 (Remarks to the Author):

This is an interesting paper that discuss the role of fetal conversion in p53 wildtype colorectal cancer cells in determining higher metastatic activity and poor prognosis. In general, the manuscript is well written and the research field is promising. English language is fine; please check throughout the text for spelling errors and consistent use of abbreviations. The aim of the study should be stated more clearly in the abstract. The Methods section is clear and exhaustive. Line 334-339: these statements should be moved to the discussion section, and the results section should only include data. The Figures and Tables are detailed and helpful for the reader. I would only suggest to include further discussion on potential clinical implications of the findings.

Reviewer #1 (Remarks to the Author):

Fetal conversion in p53 wildtype colorectal cancer cells imposes higher metastatic activity and poor prognosis Sole and co-authors.

The authors seek to demonstrate that colorectal cancers that retain wildtype p53 acquire a quiescence-like phenotype following administration of a sub-lethal dose of a clinically relevant genotoxic chemotherapeutic. They go on to demonstrate that this approach also results in induction of a fetal-like intestinal stem cell associated transcriptional program, which is dependent upon YAP1. The authors go on to show that nuclear YAP1 is associated with poor outcomes in human colorectal cancers that retain wildtype p53.

Comments to the authors –

I have a number of concerns related to this manuscript at present –

In the initial section of the manuscript “Low-dose CT treatment induces a non-senescent quiescent-like phenotype to TP53 WT cancer cell in the absence of sustained DNA damage”

1) The authors have gone to significant efforts to determine sub-lethal doses of 5-FU and Irinotecan (henceforth CT) in their collection of colorectal cancer PDOs. They go on to show that administration of these sublethal CT doses, either IC20 or IC30 for 72 hours results in a robust growth arrest which persists even following washout for 2 weeks.

a. At this point, the authors characterize in depth the growth arrested phenotype, albeit only in PDO5 (p53 WT), despite the growth arrest phenotype appearing more robust in p53 mutant lines PDO4, PDO10, PDO11 and PDO15 (although they do not mention PDO4 at this point).

Answer: As this reviewer mentions, the growth arrest phenotype is more robust in p53 mutated PDOs, which in fact acquire robust apoptotic traits as determined by analysis of Annexin V binding (new Figure S1H), and failed to restart growing after drug withdrawal (Figure S1A) and generate new organoids in the TIC assays (Figure 2B). It was the initial observation that CT-treated PDO5 showed a healthy morphology in the absence of apoptotic or senescence traits (see Figures S1G and 1E) and maintained the capacity to regrow at a very low rate after CT (Figure 1B and 1C) and to initiate new organoids when seeded as single cells (Figure 2A) (in contrast to other PDOs tested that were p53 mutants) what impelled us to investigate this phenotype. We have now included data from PDO66 and PDO20 (p53 WT) that show comparable growth arrest but consistent TIC activity after CT treatment, similar to PDO5 (new Figures S1A and S1I) and the growth curves for the p53 hypomorphic PDO4.

In the text: “Cell cycle arrest in the absence of apoptosis was observed not only in PDO5 but also in the p53 WT PDO20 and PDO66 (Figure S1I).”

We are also including new data indicating that p53 deletion (mutation) imposes defective DNA repair after sublethal CT (Figures 1K-M) and impaired the clonogenic capacity of CT-treated cells (Figures 2C and 2L).

b. The authors go on to show that IC20 and IC30 doses of CT induce substantial DNA damage in p53 mutant lines PDO4 and PDO8, but not in the p53 wildtype line PDO5. Leading to the statement that the presence of p53 permits acquisition of the PQL phenotype in the absence of DNA damage.

These observations raise a number of questions –

i) Is the phenotype ascribed to PDO5 i.e. quiescence in the absence of DNA damage shared by other p53 wildtype lines, or is it unique to this line? The authors mention PDO66 in later experiments (but share very little data related to this line) – does it exhibit the same phenotype? Are other p53 wildtype lines available to test?

Answer: We have now included data from PDO66 and PDO20 indicating that the acquisition of the quiescent phenotype after CT is not restricted to a single PDO line (new Figures S1A and S1I). Moreover, we have changed the PDO5 transcriptomic data by the combined transcriptomic analysis of PDO5 plus PDO66 (see whole data in new Table S2) and identified E2F and G2M checkpoint as the highest downregulated pathways in CT treated PDO5 and PDO66 (new Figure 3A)., indicating that cell cycle inhibition is a common trait of the two independent p53 WT PDOs analyzed.

In the text: “the E2F pathway that coordinates cell cycle progression at the G1/S transition (reviewed in (Rubin et al, 2020)) and the G2/M checkpoint were among the highest downregulated pathways, together with the MYC pathway (Figure 3A), suggesting a general inhibition of proliferation.”

We are also including data from p53-deleted PDO5 indicating that accumulation of DNA damage in these cells is p53 dependent (new Figure 1L) and γ H2A.X analysis of 2 different CRC cell lines, HCT116 (p53 WT) and DLD1 (p53 mutant), that show differential DNA damage resolution with the p53 mutant cells showing higher accumulation of DNA damage after 72 hours of treatment (Figure 1M).

ii) When looking at the DNA damage phenotype associated with p53 mutation/loss, does the PDO8 line exhibit the same growth-arrest phenotype as all other PDOs mentioned following administration of sublethal CT?

Answer: The phenotype of PDO8 is identical to other p53 mutant PDOs. Analysis of PDO8 growth after IC20-30 5-FU+Iri. treatment followed by drug washout is now included in new Figure S1A.

Why do the authors demonstrate growth arrest in PDOs 4, 5, 10, 11 and 15, but assess DNA damage phenotype in PDOs 4, 5 and 8?

Answer: We just selected these p53 mutant PDOs as representative because they were in culture when performing these experiments. However, the same DNA damage phenotype is observed in all tested p53-deficient PDOs and CRC cell lines as well as in the p53-deleted PDO5, which is not directly related with the 5-FU+Iri. doses. We are now including growth curves from PDO8 (Figure S1A) and DNA damage data from p53 WT and KO PDO5 (Figure 1K and 1L) and in two different p53 mutant CRC cell lines treated at the same doses of 5-FU+Iri. (Figure 1M).

iii) P53 loss or mutation is associated with tolerance of increased or sustained DNA damage, and is associated with resistance to genotoxic CT regimens. This is reflected by the absolute dose of CT administered as IC20 + IC30 for the individual lines studied, where for example the IC20 values for PDO4 were 1.25 ug/ml 5-FU and 0.5 ug/ml Irinotecan, while for PDO5 these were 0.14 ug/ml 5-FU and 0.06 ug/ml irinotecan (roughly 10x lower for each). It is conceivable that observed DNA damage phenotype observed in the p53 mutant lines arises solely because they are able to survive significantly higher levels of DNA damage driven by higher doses of CT, while the p53 WT lines are not. The phenotype could be that the IC20 of CT in PDO5 is at a level below that required to elicit significant DNA damage, but high enough to drive quiescence. In order to

demonstrate that the adoption of a quiescent state in the absence of increased/sustained DNA damage is a unique feature of p53 WT cells, the authors should assess both DNA damage phenotypes and quiescence of p53 mutant/deficient lines at lower doses of CT (eg the IC20 of PDO5).

Answer: We thank the reviewer for addressing this important issue that we had not properly focused throughout the manuscript. It was recurrently shown that p53 deficiency results in a superior capacity to resist chemotherapy by different mechanisms (reviewed in Michel et al., Cancers 2021) and this is also exemplified by the higher concentrations of 5-FU+Iri. that are required to achieve the IC20 and IC30 in most p53 mutant PDOs and cell lines (new Table S1). However, as suggested by the reviewer, we are now including data from p53 KO PDO5 treated at the same 5-FU+Iri. doses as the WT (IC30 for PDO5 WT that represents IC20 for the KO) (new Figures 1L, 2C and 5G), PDO66 (p53 WT) that show CT sensitivity comparable to p53 mutants (new Figures 2A and S1I) and HCT116 (p53 WT) and DLD1 (p53 mutant) treated at the same 5-FU+Iri. doses (new Figure 1M).

Together, our results indicate that 5-FU+Iri. treatment at the minimum doses producing detectable effect on cell growth after 72 hours imposes a quiescent/no-damaged phenotype that is preferentially observed in the context of functional p53. In contrast, p53 mutant or p53-deleted cells accumulate massive amount of DNA damage and fail to propagate new tumors both in vitro (TIC assays in Figure 2B and new Figure 2C) and in vivo (new Figure 2L). Moreover, differences in CT dosage does not explain accumulation of DNA damage (Figures 1K-M) and defective clonogenic activity (Figure 2C) that are specifically observed in p53 deficient cells after sublethal CT. We have now clarified this concept in the discussion section: "Although, it is still possible that p53 mutated cells can acquire a PQL phenotype at specific CT doses, we found that p53 deletion by itself led to accumulation of DNA damage and imposed defective clonogenic activity after exposure to comparable levels of 5-FU+Iri."

iv) The question raised in iii) above could be partially addressed in Figure 1K, where the authors demonstrate induction of DNA damage reflected by γ H2A.X induction following P53 deletion in PDO5, but this lacks clarity. It is notable that the sublethal CT treatment here does not increase γ H2A.X above the already high basal level, in contrast to the p53 mutant lines discussed previously (e.g PDO4, PDO8; Fig S1H).

Answer: We are aware that p53-depleted PDO5 displayed basal γ H2A.X levels that are much higher than all other p53 mutant cells, which made difficult detecting the effects of CT. One possible explanation is that p53 deletion produces a more severe phenotype than p53 mutations that differently impact on its activity or, alternatively, that the DNA-damaging phenotype of p53-loss-of-function is attenuated with time and passages.

To better show the effects of p53 deletion in PDO5 cells, we are now including comet assays of p53 WT and KO PDO5 that show sustained DNA damage in the latter even when treated at the same 5-FU+Iri. doses (new Figure 1L). Moreover, we have changed the previous WB analysis of PDO5 WT and p53 KO cells at 72 hours treatment by time-course experiments. We found that sublethal CT in p53 KO PDO5 imposed a temporary increase of γ H2A.X levels from 6 hours to 48 hours that is then attenuated at 72 hours (new Figure 1K). Importantly, p53 KO PDO5 showed massive DNA damage (compared with the parental PDO5), which is linked with a reduction in tumor-initiating-capacity (new Figure 2C).

Is this dose given to the PDO5 p53 KO line at the IC20 generated for the parental PDO5 line, or is it a newly generated IC20 for this specific P53 KO line?

Answer: We have determined the IC20-30 for p53 KO PDO5 and found that p53-deficient cells are a little more resistant than parental PDO5 (see new table S1). However, in all experiments comparing PDO5 WT and KO we have treated cells with the same 5-FU+Iri. doses that result in IC30 for the WT and IC20 for the p53 KO.

I think this points to the suggestion in point iii) above that the IC20 dose for PDO5 does not induce a strong DNA damage response under any circumstances.

Answer: We agree with the reviewer that p53 mutant cancer cells (at least some of them) can be insensitive to CT doses that impose IC20-30 to PDO5, which is in agreement with the notion that p53 mutations may increase CT refractoriness (reviewed in Michel et al., Cancers 2021). However, we now demonstrate that p53-deleted PDO5 cells and p53 mutant CRC cell lines treated at the same doses as their p53 WT counterparts show higher amounts of basal and CT-induced DNA damage (new Figures 1K, 1L and 5G) and defective new TIC (Figure 2C). We are also showing that PDO66, which show CT sensitivity comparable to most p53 mutant PDOs used in this study (5-FU:2.50, Iri:1.00 compared with 5-FU:1.56, Iri:0.63 for PDO8, and 5-FU:2.00, Iri:0.80 for PDO4) show proliferation arrest after treatment in the absence of cell death (new Figure S1I), partially restores its growing capacity after drug withdrawal (new Figure S1A) and preserves a robust clonogenic activity in TIC assays after IC20-30 treatment (new Table S1 and Figure 2A). Our conclusion (now further explained in the discussion section) is that CT doses that are sublethal for a particular CRC tumor (that will vary from cell to cell) differentially impact on p53 WT and mutant cells, being the PQL phenotype specifically acquired in the presence of functional p53.

In the text: "Notably, cancer cells carrying dysfunctional p53 show partial conversion to fetal phenotype but fail to acquire a functional PQL phenotype, as they lose TIC associated with massive accumulation of DNA damage, when treated at doses that reduce cell growth about 20-30 %. Although, it is still possible that p53 mutated cells can acquire a PQL phenotype at specific CT doses, we found that p53 deletion by itself led to accumulation of DNA damage and imposed defective clonogenic activity after exposure to comparable levels of 5-FU+Iri."

As a minor point, why in the control lane (PDO5, untreated) for this panel is P53 detectable, in contrast eg to Figure 4a?

Answer: We apologize for this apparent lack of consistency. However, we have repeatedly observed that p53 detection can vary from experiment to experiment, and also depending on the amount of protein that is loaded, which is difficult to standardize when working with PDOs.

This section does not convince me that the DNA damage phenotype in the p53 mutant lines is linked to the PQL phenotype (or indeed the absence of DNA damage in the p53 wildtype line), and indeed therefore that the presence of wildtype p53 is the key factor. This has an impact for the remainder of the study – for example the TIC experiments in the next section.

Answer: We agree with the reviewer that we cannot directly link the massive increase in DNA damage observed in p53 mutant CRC cells (Figures 1J, 1L, S1J and S1K) with their loss of clonogenic activity both in vitro and in vivo, although this would be a reasonable connection. However, new experiments including additional p53 WT PDO cells (new Figure 2A and S2A), experiments with p53 KO PDO5 treated at the same CT doses as parental PDO5 (new Figures 2C and 5G) and in vivo experiments with PDO4 and PDO8 (new Figure 2L) should help to convince the reviewer that p53 is a

key factor in the acquisition of this phenotype. Moreover, the combined RNA-seq analysis of PDO5 and PDO66 (new Table 2 and new Figure 3A) also supports the concept p53 WT PDO cells (not just PDO5) acquire a PQL after sublethal CT. Nevertheless, the possibility that p53-deficient PDOs could share the capacity to acquire a PQL phenotype after damage would not reduce the scientific impact of our findings.

In the second section of the manuscript, entitled “PQL cancer cells display increased in vitro and in vivo tumor initiation capacity” -

1) I have some concern about the description of the use of PDO66 in this section of the manuscript (and elsewhere), primarily that the authors cite data related to PDO66, but do not include any data in the either the primary figure or supplementary to support their findings (related to figure 2A). Moreover, we do not have any data from the earlier section to suggest that PDO66 exhibits the same phenotype in terms of quiescence (the “PQL” phenotype), and absence of DNA damage as described for PDO5. This is a key point.

Answer: Following the reviewer suggestion, we have now included IHC analysis of ki67 and cleaved-Caspase 3 from PDO20 and PDO66 (new Figure S1I) and growth curves after drug withdrawal (new Figure S1A) indicating a comparable PQL phenotype after CT in the different p53 WT PDOs tested. To avoid increasing the number of panels in Figure 3, we have now combined RNA-seq data from both PDO5 and PDO66 (removing the batch effect, see methods) and performed all analysis with genes that are differentially expressed in both CT-treated organoids. In new Figure 3, we have now changed results from PDO5 by results obtained by the in-common analysis of PDO5 and PDO66 RNA-seq (new Figures 3A, 3D-G and Supplementary Table S2). We have also included data showing that the top downregulated genes in CT-treated PDOs are MYC and E2F targets and G2M checkpoint elements (new Figure 3A) further supporting the concept that sublethal CT results in proliferation inhibition.

In the text: “Cell cycle arrest in the absence of apoptosis was observed not only in PDO5 but also in the p53 WT PDO20 and PDO66 (Figure S1I).” and “the E2F pathway that coordinates cell cycle progression at the G1/S transition (reviewed in (Rubin et al, 2020)) and the G2/M checkpoint were among the highest downregulated pathways, together with the MYC pathway (Figure 3A), suggesting a general inhibition of proliferation.”

We are also including here the analysis of PDO66 alone (Figure below) to show that PDO5 and PDO66 acquire an almost identical phenotype after sublethal CT (even when they show very different CT sensitivity).

2) The intracardiac injection of PDO5 following administration of CT suggests an interesting phenotype, suggesting that sublethal CT does induce a metastatic phenotype/survival advantage to PDO5, although unfortunately appears underpowered. It would also be useful to know whether p53 mutant/deficient lines behave in the same way.

Answer: As mentioned by the reviewer, differences in the intracardiac assays were not significant, which could be partially explained by the fact that this assay do not reflect all stages of the metastatic process such as EMT or invasive capacity (note that EMT is one of the pathways enriched in CT treated PDOs). However, we have now performed new intracardiac assays to increase significance (new Figure 2E) and to determine the possibility that CT-treatment favors the initial metastatic colonization capacity of PDO cells (new Figure 2F). We found a dose-dependent although non-significant trend of CT-treated cells for metastatic outgrowth ($p=0.10$) and a significant difference in metastasis initiation (new Figure 2F).

In the text: "We found that PDO5 treated with 5-FU+Iri. displayed a superior and dose-dependent, although non-significant, metastatic capacity than untreated cells (logistic regression trend test, $p=0.108$). Specifically, 7 of 14 mice transplanted with untreated PDO5 cells showed visible metastasis 15 weeks after injection compared with 4 of 6 mice transplanted with IC₂₀-treated cells and 9 of 11 mice with IC₃₀-treated cells (Figure 2D and 2E). Quantitative analysis of the evolution of lesions in an independent assay demonstrated a significant higher capacity of IC₃₀-treated cells for metastasis initiation (Figure 2F)."

We have also performed the suggested experiments using p53 mutant PDOs by intracecal transplantation of tumor cells. These results are now included in new Figure 2L and confirmed the defective clonogenic capacity of CT-treated p53 mutant cells observed in vitro (Figures 2B and 2C).

In the text: "Parallel in vivo experiments comparing IC₂₀-treated PDO5, PDO4 and PDO8 cells indicated a defective capacity of p53 mutant cells to generate in situ tumors and intraperitoneal implants after sublethal CT treatment (Figure 2L), which was in agreement with their defective TIC in the in vitro assays."

In the discussion section we now mention the differences between metastatic capacity of sublethal CT treated cells observed in the intracardiac and the intracecal assays: "Of note, whereas differences in tumor dissemination between untreated and CT-treated PDO5 in the intra-cecal transplantation experiments were highly significant, we only detected slight differences in the intracardiac assays. These divergent results were likely indicating that intracardiac injection does not reflect all steps of the metastatic process such as EMT or invasive capacity. Supporting this possibility, EMT is one of the pathways that are enriched in the CT treated PDOs (see Figure 3A), and we have now confirmed that EMT genes are specifically upregulated in the quiescent cell population (see Figure 3H)."

3) The section related to cecal transplantation is confusing – the text states that "IC20 and IC30-treated derived tumors significantly smaller than those arising from untreated controls (Figure 2F)" – but Figure 2F relates to number rather than size. Should this be Figure 2G, a measure of inoculation site vs peritoneal dissemination?

Answer: We apologize as we involuntary changed figures 2F and 2G. We have now corrected this.

In figure 2G, it is not clear what the tumor weight measurement actually refers to – is it the average cumulative weight of all IP implants found within each mouse?

Answer: We represented the total weigh of implants per animal. We have now included this information in the corresponding figure legend.

As previously, while the phenotype observed under the conditions tested are absolutely correct, it is not fair to suggest that they are representative of P53 wildtype colorectal cancers as a whole, and may equally be a product of experimental approach taken.

Answer: We totally agree with the referee's observation. To avoid suggesting that our results are representative of all p53 WT tumors, we have rephrased the text to: "These results indicate that TP53 WT PDO5 cells treated with sublethal 5-FU+Iri. show reduced capacity to proliferate in vitro and in the primary tumors, but display comparable TIC as untreated cells in vitro and higher metastatic activity in vivo"

Section entitled "CT-induced PQL cells display fetal intestinal stem cells (feISC) characteristics"-

The induction of fetal/regenerative stem cell like signatures following sublethal CT in this setting is very interesting and may be indicative of regeneration/recovery following loss of the Lgr5 canonical/homeostatic stem cell population. It is great to note that the observations related to PDO5 seem to be supported by data generated from PDO66, although again this must be fully presented for the reader rather than stated in the text – specifically the shift towards fetal phenotype in PDO66 must be shown rather than "not depicted".

Answer: We are now including RNA-seq data from both PDO5 and PDO66 in new Table S2 and substituted all the analysis from PDO5 RNA-seq DEG in Figure 3 by the combined PDO5 plus PDO66 RNA-seq data analysis (removing the batch effect as it is mentioned in methods). Our results indicate a comparable fetal conversion of CT-treated PDO5 and PDO66 cells in this analysis (new Figures 3D-G).

In the text: "our analysis identified an inversed correlation between DEG in PDO5 and PDO66 and the canonical Lgr5+ ISC signature (Figure 3D)". See also analysis from PDO66 included in this rebuttal letter.

If the association of this phenotype and earlier sections are to stand up, it is important to demonstrate here that this fetal regenerative transcriptional signature is not induced in p53 mutant or deficient line eg PDO4, 8 or the PDO5 p53 KO line (I note that this has been attempted in later sections). It is also important here to test both at the IC20/30 of the p53 deficient/mutant lines, and at the dose given to PDO5.

Answer: From the results obtained with the different models, we cannot conclude that the fetal signature is specific of p53 WT cells, and in fact, we show that several of the genes tested are also induced in p53 mutant cells treated at IC20-30 although at different levels (Figures 5E-H). Our interpretation is that p53 mutant or KO cells can initially respond to CT by activating fetal genes, however the capacity to efficiently repair DNA damage is p53 and p21 dependent (Figures 1I-L and 5G). We are now including data from PDO5 p53 WT and KO treated at the same CT doses and PDO66 (p53 WT) demonstrating that p53 WT and mutant cells respond differently after sublethal CT, which is not dependent on the concentration of 5-FU+Iri. that is given. Moreover, the likely possibility p53 mutant cells could be insensitive to the doses that produce a 20-30% reduction of PDO5 growth would not contradict the message of our work.

We are now clarifying this concept in the discussion section: "Notably, cancer cells carrying dysfunctional p53 show partial conversion to fetal phenotype but fail to acquire

a functional PQL phenotype, as they lose TIC associated with massive accumulation of DNA damage, when treated at doses that reduce cell growth about 20-30 %. Although, it is still possible that p53 mutated cells can acquire a PQL phenotype at specific CT doses, we found that p53 deletion by itself led to accumulation of DNA damage and imposed defective clonogenic activity after exposure to comparable levels of 5-FU+Iri.”

Section entitled “Acquisition of FeISC by CT treatments in linked to and dependent upon YAP1 activation” –

Putting aside my concerns around the use of a single organoid line, and the link between p53 status and the observed phenotype, this section has a number of key, and very important observations – critically the induction of nuclear YAP1 and fetal-associated markers in human CRC samples post neoadjuvant CT, when compared to paired pre-treatment biopsies. Although the link to “sublethal CT” in line with the in vitro data generated in the manuscript is tenuous.

Answer: We thank the reviewer for the positive comment, although we do not totally understand the sentence “link to sublethal CT in line with the in vitro data generated in the manuscript is tenuous”. Because cancer patients are never voluntarily treated with sublethal CT, we considered neoadjuvant-treated tumors displaying incomplete remission as the closest patient model to validate our experimental data generated using sublethal CT.

Analysis of this cohort of samples indicated that human CRC tumors displaying partial remission (sublethal CT) show accumulation of nuclear YAP1 and fetal gene expression, comparable to the results obtained from in vitro treatment of PDOs.

In the text: “the number of tumors carrying nuclear YAP1 was massively increased after neoadjuvant treatment (35 out of 45 analyzed) (Figure 4E and Supplementary Table S3), which was associated with expression of the feISC markers S100A4 and SERPINH1 (Figure 4F).”

Next, because we already detected nuclear YAP1 in few untreated tumors, we analyzed a larger cohort of untreated samples, and found a significant association between nuclear YAP1 accumulation and patient prognosis. Moreover, we have now included data indicating that YAP1 deletion sensitizes PDO5 cells to CT (new Table S1), precludes their clonogenic activity (new Figure 4K) and results in DNA damage accumulation after CT (new Figure 4L).

In the text: “Considering the mean value \pm 0.2 standard deviations of the H-score, we observed a trend towards poor prognosis in the group with higher nuclear YAP1 (disease-free survival: $p=0.26$; $HR=1.38$) that increased when considering mean value \pm 0.4 s.d. ($p=0.12$; $HR=1.58$). Prognosis value of nuclear YAP1 levels reached statistical significance when considering the mean value \pm 0.6 standard deviations ($p=0.039$; $HR=1.97$) (Figure 4H and 4I). Indicative of the clinical applicability of our observations, verteporfin treatment increased the sensitivity of PDO5 cells to 5FU+Iri. (Figure 4J). Further suggesting that verteporfin effects are linked to YAP1 inhibition, genetic YAP1 deletion precluded PDO5 clonogenicity in TIC assays (Figure 4K) associated with increased basal and CT-induced DNA damage (Figure 4L).”

We believe that these results (even if they have been obtained from a limited cohort of patients) are clinically relevant and have to be included in the manuscript.

Section entitled – “A restricted YAP1-dependent fetal signature shows coordinate expression in human CRC associated with higher p21 levels”

This section really nicely describes the generation of a canonical signature associated with the feISC population, and subsequent use across multiple human cohorts. Again, where a study has been carried, I would ask that the

authors include the data, in this case in reference to the association between p21 in the Jorissen and TCGA datasets.

Answer: We have included the 28up+8down-feISC signature data from the Jorissen and TCGA datasets (new Figure S5B) and its association with p21 levels (new Figure S5C).

The correlation of the feISC signature generated here with the CM4 (and anti-correlation with CMS2) from the Guinney et al 2015 paper is very interesting, and demonstrates really nicely that the fetal program may be associated with more aggressive disease (and a poorer prognosis). Given that around 50% of patients in CMS2 and CMS4 groups from the Guinney et al dataset exhibit p53 mutation, it would be very interesting to know how closely the 28up+8down signature aligns to p53 status in these groups.

Answer: This is in fact an interesting observation. We have now analyzed the possibility that feISC is linked to the TP53 status in the different molecular subtypes, and in particular in CMS4. We observed a slight but consistent accumulation of TP53 WT in tumors carrying the feISC signature in this molecular subtype. However, what are the functional implications of this association should be further studied.

These results are now included in new Figure S6D and in the text: “Interestingly, we observed a slight accumulation of TP53 WT in tumors carrying the 28up+8down-feISC signature in the subgroups of CMS4 from TCGA and Marisa cohorts (Figure S6D).”

Section entitled – “The Yap-dependent feISC signature is predictive of reduced disease-free survival in TP53 WT tumors”

As with the previous section, this data is really nice – it demonstrates clearly the prognostic value of YAP1 dependent feISC signatures in human CRC. While the authors take encouragement from the link between TP53 status, the 28up-8down signature and prognosis in stage II+III patients, it is unusual to focus on this, when the data indicates that the 28up-8down signature is a good prognosticator in unselected patient populations in all datasets studied – i.e it is not selective for p53 status.

Answer: We agree with the reviewer that the 28up-8down signature is sufficient to predict prognosis in the whole patient population and we very clearly show this. However, because our data suggest a link between p53 and acquisition of the PQL phenotype, we tested whether 28up-8down showed any difference in its prognosis capacity of p53 WT or mutated tumors. Surprisingly, prognosis value of the 28up-8down signature was primarily gathered in the p53 WT tumors, as p53 mutations seem to impose a prevalent effect over fetal conversion. Thus, we consider important including the concept of p53 status as it could be useful for future studies or for patient stratification in clinical trials involving YAP1 inhibitors.

Reviewer #2 (Remarks to the Author):

In this study Sole and colleagues investigate the role of Yap dependent fetal conversion in mediating tumor cell therapy resistance. They find that in P53 wild-type colorectal tumor organoids treatment with a sub-lethal dose of chemotherapy induces a quiescent, fetal-like state on tumor cells. These cells have high levels of Yap activity and show some evidence of increased metastatic potential. They further show that tumor cells with these properties are found in patient samples and are predictive of poor disease outcome. The concepts described here are interesting and could potentially indicate a mechanism to target therapy resistant colorectal cancer cells via targeting the Yap pathway. Although the role of Yap in fetal conversion and the induction of quiescent like phenotypes following therapy have been previously described the linking of these states appears novel, timely and solidifies the evidence of the importance of these processes. However, a number of the key conclusions of the manuscript require further strengthening prior to publication, in particular the role of these cells in metastasis and the potential role of Yap1 in this process.

1) One of the key findings of the paper is that sub-lethal chemotherapy doses tumor cells have increased tumorigenicity in vivo and increased metastatic potential (Figure 2). However, the first of these experiments did not show a statistically significant difference (2C-E) negating the importance of this finding.

Answer: The reviewer is correct that differences of tumorigenicity in the intracardiac assays were not statistically significant, whereas differences in tumor implant generation in the intra-cecal PDO transplantation were much higher and highly significant. These divergent results were likely indicating that intracardiac injection does not reflect all steps of the metastatic process such as EMT or invasive capacity. Supporting this possibility, EMT is one of the pathways that are enriched in the CT treated PDOs (Figure 3A), and we have now confirmed that EMT genes are specifically upregulated in the quiescent cell population (new Figure 3H). However, we have now performed additional intracardiac assays to increase statistical significance and we found a dose-dependent trend towards higher metastatic activity in CT-treated PDO5 ($p=0.10$) (new Figure 2E) and a significant increase in metastasis initiation (new Figure 2F).

In the text: "We found that PDO5 treated with 5-FU+Iri. displayed a superior and dose-dependent, although non-significant, metastatic capacity than untreated cells (logistic regression trend test, $p=0.108$). Specifically, 7 of 14 mice transplanted with untreated PDO5 cells showed visible metastasis 15 weeks after injection compared with 4 of 6 mice transplanted with IC₂₀-treated cells and 9 of 11 mice with IC₃₀-treated cells (Figure 2D and 2E). Quantitative analysis of the evolution of lesions in an independent assay demonstrated a significant higher capacity of IC₃₀-treated cells for metastasis initiation (Figure 2F)."

We are also discussing the differences obtained in the intracardiac and intra-cecal assays in the discussion section: "Of note, whereas differences in tumor dissemination between untreated and CT-treated PDO5 in the intra-cecal transplantation experiments were highly significant, we only detected slight differences in the intracardiac assays. These divergent results were likely indicating that intracardiac injection does not reflect all steps of the metastatic process such as EMT or invasive capacity. Supporting this possibility, EMT is one of the pathways that are enriched in the CT treated PDOs (see Figure 3A), and we have now confirmed that EMT genes are specifically upregulated in the quiescent cell population (see Figure 3H)."

In addition, it appears as though bulk PDO cells have been used for this experiment rather than the quiescent population in question. Thus, it is not clear which cells following treatment have metastatic capacity. The authors should

repeat these experiments with quiescent cells using the H2B-GFP strategy employed for other experiments.

Answer: The experiments proposed by the reviewer are extremely difficult to carry out because of the very low number of viable H2B-GFP-high cells that we recover after organoid disaggregation and cell sorting. For this reason, we had initially addressed this particular issue by in vitro TIC assays (Figure S2B and S2C) and we are now including data indicating that canonical EMT genes such as SNAI2 are specifically retained in the quiescent population upon CT (new Figure 3H). However, we have now performed the suggested experiment by inoculating 5000 purified GFP-high or low cells per animal in 5 animals each. We found that even with this low number of cells, the sorted GFP-high (quiescent) population show increased TIC in vivo (3 of 5 mice with tumors compared with 1 of 5 mice transplanted with GFP-low).

In the text: "By transplantation of 5000 GFP_{high} or GFP_{low} PDO5 sorted cells in the cecum of nude mice, we confirmed the higher clonogenic activity of the quiescent population in vivo. In particular, 3 out of mice transplanted with GFP_{high} cells developed tumor in the cecum and/or intraperitoneal implants compared with 1 out of 5 mice transplanted with GFP_{low} cells after 2 months (Figure S2D)."

2) In addition to this, the role of Yap1 in imparting the fetal-like transcriptional signature is investigated but functional analysis of how this relates to tumor phenotypes is lacking. The authors should utilize Yap1 KO PDO lines to determine whether this affects the tumor initiating and/or metastatic capacity of sub-lethal dosed PDO cells.

Answer: We have now further analyzed the phenotype of YAP1 KO PDO5 and detected a slight increase in CT sensitivity compared with its WT counterpart (supplementary Table S1), which parallels the effect of verteporfin treatment (Figure 4J). Moreover, YAP1 KO PDO5 cells treated at the same 5-FU+Iri. doses as the parental PDO5 show a significant reduction in tumor initiating capacity (in vitro) when compared to the YAP1 WT (new Figure 4K) and increased DNA damage upon 5-FU+Iri. exposure (new Figure 4L).

In the text: "Further suggesting that verteporfin effects are linked to YAP1 inhibition, genetic YAP1 deletion precluded PDO5 clonogenicity in TIC assays (Figure 4K) associated with increased basal and CT-induced DNA damage (Figure 4L)."

3) The relative role of the quiescent vs non quiescent cells in PDOs following sub-lethal chemotherapy is not completely clear. The authors carry out an in vitro experiment demonstrating similar colony forming capacity but some additional characterization would be useful. Are there differences in Yap activity / marker gene expression for example?

Are there any phenotypic differences in culture other than proliferation? Have they preferentially induced an EMT like programme?

Answer: Since we observed that 90% of PDO5 cells were ki67 negative (Figure 1D) at 72 hours after treatment and this phenotype was primarily maintained 1-2 weeks after drug washout (Figure 1B and S1B), we had considered that the majority of changes observed in the bulk population inside this period (i.e. activation of the fetal-ISC and the EMT programs) could be ascribed to the quiescent cells. Nevertheless, we have now directly addressed this issue by qPCR analysis of sorted H2B-GFP high (quiescent) and GFP low (proliferating) 5-FU+Iri.-treated PDO5 cells. We found that genes contributing to feISC (Yap1-dependent) and EMT pathways in the DEG analysis of Figure 3A are significantly (massively) increased in the quiescent (GFP high) population induced by sublethal CT but primarily downregulated in the proliferating (GFP low) cells (new Figure 3H).

In the text: “We used the PDO5-hG line to test whether upregulation of genes contributing to fEISC and EMT pathways upon sublethal CT was present in the quiescent cell population. Cells were treated as explained before (see Figure S2B), sorted based on GFP levels and processed for qPCR analysis. We detected a massive upregulation of fEISC and EMT genes in CT-treated cells that was restricted to the GFP_{high} population (Figure 3H).”

4) The phenotypes observed appear to be specific for P53 wild-type tumors but the mechanism isn't explored. One possible explanation raised by the authors is that P53 allows survival of the Yap activated cells. The authors could investigate this possibility by analyzing the P53 KO PDO5 model over a time course for Yap activity and apoptosis induction.

Answer: Our model, that we tried to better explain in the manuscript is that YAP1, induced by damage, imposes the activation of a transcriptional program (Figures 4C and 4D) associated with malignant traits such as higher metastatic capacity (Figures 2G, 2H, 3A and 3H), and is predictive of poor prognosis in human CRC (Figure 6). In contrast, p53 through p21 is primarily involved in inducing a quiescent/non-proliferative phenotype after sublethal CT treatment (that is concomitant with YAP1 activation) that may facilitate DNA repair and cells survival, as suggested by the high accumulation of DNA damage in the p53 mutant or deleted cells (new Figures 1K and 1L, and S1H, S1J and S1K) even when treated as the same 5-FU+Iri. doses (new Figures 1K and 1L). We have also performed time-course assay of PDO5 p53 KO and WT treated with low-dose CT followed by western blot analysis showing that YAP1 and downstream fetal markers such as TIMP2, MRAS and ICAM are similarly induced at early times of CT treatment in the absence of functional p53. However, p53 KO display a massive amount of damage as determined by gH2A.X (new Figure 1K) and apoptosis measured by cleaved Caspase 3 and cleaved PARP1 levels in these same experiments (new Figure 5G).

Minor points:

1) There is a lack of quantification throughout the manuscript. This is particularly apparent in Figure 1 where the key findings of the quiescent cell population are described. Quantification and statistical analysis of 1B, 1C, 1E, 1F and 1H should be included.

Answer: We have added statistics in 1B, 1C and 1F and have quantified the western blot analysis in Figure 1H. In Figure E, we have added the average value, deviation and significance from 3 biological replicates performed.

2) Same as above for Figure 4B – quantification of Yap1 nuclear localization % should be included.

Answer: We have included quantification of nuclear YAP1 and statistics in new Figure 4B.

Reviewer #3 (Remarks to the Author):

This is an interesting paper that discuss the role of fetal conversion in p53 wildtype colorectal cancer cells in determining higher metastatic activity and poor prognosis. In general, the manuscript is well written and the research field is promising. English language is fine; please check throughout the text for spelling errors and consistent use of abbreviations. The aim of the study should be stated more clearly in the abstract. The Methods section is clear and exhaustive. Line 334-339: these statements should be moved to the discussion section, and the results section should only include data. The Figures and Tables are detailed and helpful for the reader. I would only suggest to include further discussion on potential clinical implications of the findings.

Answer: We really appreciate the positive comments from the reviewer. We have now included all suggestions from the reviewer in this revised version.

REVIEWER COMMENTS

Reviewer #1 (Remarks to the Author):

I would like to thank the authors for their significant effort in addressing my concerns - i think the manuscript is greatly improved, and remains an important study.

My only remaining concern relates to the RNAseq dataset which now makes up Fig 3A / Supplementary Table S2. The authors have presented a combined analysis of PDO20 and PDO66, albeit with batch correction (NB spelling error in title of Supplementary Table S2), with GSEA analysis indicating a positive enrichment of e.g p53 pathway and negative enrichment of e.g. Myc targets, E2F targets and G2M checkpoint upon treatment. Despite the batch correction, presenting in this way does not give a clear indication of the variation in response between the 2 independent PDOs. Could the authors please rectify this, possibly using one of the following approaches -

- 1) Perform the analysis as single-sample GSEA on all samples with using the current genesets, this could then be presented as a normalised heatmap (would give the reader a clearer picture of the variance across the different lines)
- 2) Perform the GSEA analysis in PDO66 and PDO20 independently and present both datasets – this approach would hopefully allow each dataset to be corroborated. I note that the data for PDO66 was presented as a reviewer figure, but my concern here is that the Myc targets, E2F targets and G2M target genesets were not shown in this figure (are they negatively enriched in this line?)
- 3) Provide a PCA (or equivalent) of all samples to demonstrate that the response to treatment in PDO20 and PDO66 are similar/equivalent.

My preference would be that the authors perform the analysis outlined in either 1 or 2 above, but also include a PCA plot (3) as a supplementary panel.

If the authors could address this concern, I would be more than happy to see this important study published in Nature Communications.

Reviewer #2 (Remarks to the Author):

The authors have addressed my previous concerns and I now recommend publication of this study.

In Barcelona, March 14, 2022

Reviewer 1 comments:

My only remaining concern relates to the RNAseq dataset which now makes up Fig 3A / Supplementary Table S2. The authors have presented a combined analysis of PDO20 and PDO66, albeit with batch correction (NB spelling error in title of Supplementary Table S2), with GSEA analysis indicating a positive enrichment of e.g p53 pathway and negative enrichment of e.g. Myc targets, E2F targets and G2M checkpoint upon treatment. Despite the batch correction, presenting in this way does not give a clear indication of the variation in response between the 2 independent PDOs. Could the authors please rectify this, possibly using one of the following approaches-

- 1) Perform the analysis as single-sample GSEA on all samples with using the current genesets, this could then be presented as a normalised heatmap (would give the reader a clearer picture of the variance across the different lines).
- 2) Perform the GSEA analysis in PDO66 and PDO20 independently and present both datasets – this approach would hopefully allow each dataset to be corroborated. I note that the data for PDO66 was presented as a reviewer figure, but my concern here is that the Myc targets, E2F targets and G2M target genesets were not shown in this figure (are they negatively enriched in this line?)
- 3) Provide a PCA (or equivalent) of all samples to demonstrate that the response to treatment in PDO20 and PDO66 are similar/equivalent.

My preference would be that the authors perform the analysis outlined in either 1 or 2 above, but also include a PCA plot (3) as a supplementary panel.

Answer: I would like to clarify that the data presented was generated from PDO5 and PDO66 (instead of PDO20). We have now changed Figure 3 based on your suggestions and preferences, and have modified the text accordingly:

“Principal Component Analysis (PCA) indicated that untreated PDO5 and PDO66 clustered together and formed separate entities when compared with IC20- and IC30-treated PDO5 or PDO66 cells (Figure 3A). Gene Set Enrichment Analysis (GSEA) of genes differentially expressed in CT-treated PDO5 (Figure 3B) and PDO66 (Figure 3C) uncovered p53 as the main activated pathway in CT-treated cells, which was confirmed in PDO5 by WB analysis (Figure 3D)”

We have also corrected the mistake in the title of Table S2.

Thank you very much for helping us to improve our work.

Sincerely,
Lluís Espinosa